# Regional background $O_3$ and $NO_x$ in the Houston-Galveston-Brazoria (TX) region: A decadal-scale perspective

Loredana G. Suciu[1], Robert J. Griffin[2], Caroline A. Masiello[1]

[1]Department of Earth Science, Rice University, Houston, 77005, USA

[2]Department of Civil and Environmental Engineering, Rice University, Houston, 77005, USA

*Correspondence to*: Loredana G. Suciu (lgs4@rice.edu)

**Abstract.** Ozone ($O_3$) in the lower troposphere is harmful to people and plants, particularly during summer, when photochemistry is most active and higher temperatures favor local chemistry. Local precursor emissions, such as those of volatile organic compounds (VOCs) and nitrogen oxides ($NO_x$), together with their chemistry contribute to the $O_3$ and $NO_x$

mixing ratios in the Houston-Galveston-Brazoria (HGB) region. In addition to local emissions, chemistry and transport, larger-scale factors also contribute to local $O_3$ and $NO_x$. These additional contributions (often referred to as "regional background") are not well quantified within the HGB region, impeding more efficient controls on precursor emissions to achieve compliance with the National Ambient Air Quality Standards for $O_3$. In this study, we estimate ground-level regional background $O_3$ and $NO_x$ in the HGB region and quantify their decadal-scale trends.

We use four different approaches based on principal component analysis (PCA) to quantify background $O_3$ and $NO_x$. Three of these approaches consist of independent PCA on both $O_3$ and $NO_x$ for both 1-h and 8-h levels to compare our results with previous studies and to highlight the effect of both temporal and spatial scales. In the fourth approach, we co-varied $O_3$, $NO_x$ and meteorology.

Our results show that the estimation of regional background $O_3$ has less inherent uncertainty when it was constrained by $NO_x$

and meteorology, yielding a statistically significant temporal trend of -0.68 ± 0.27 ppb $y^{-1}$. Likewise, the estimation of regional background $NO_x$ trend constrained by $O_3$ and meteorology was -0.04 ± 0.02 ppb $y^{-1}$ (upper bound) and -0.03 ± 0.01 ppb $y^{-1}$ (lower bound). Our best estimates of 17-y average of season-scale background $O_3$ and $NO_x$ were 46.72 ± 2.08 ppb and 6.80 ± 0.13 ppb (upper bound) or 4.45 ± 0.08 ppb (lower bound), respectively. Average background $O_3$ is consistent with previous studies and between the approaches used in this study, although the approaches based on 8-h averages likely

overestimate background $O_3$ compared to the hourly median approach by 7-9 ppb. Similarly, the upper bound of average background $NO_x$ is consistent between approaches in this study (A-C), but overestimated compared to the hourly approach by 1 ppb, on average. We likely overestimate the upper bound background $NO_x$ due to instrument overdetection of $NO_x$ and the 8-h averaging of $NO_x$ and meteorology coinciding with MDA8 $O_3$.

Regional background $O_3$ and $NO_x$ in the HGB region both have declined over the past two decades. This decline became

steadier after 2007, overlapping with the effects of controlling precursor emissions and a prevailing southeasterly-southerly flow.

**Keywords:** tropospheric ozone, nitrogen oxides, regional background, covariance of chemistry and meteorology, temporal trends

# 1 Introduction

In the lower troposphere, ozone ($O_3$) has impacts on both human health and ecosystems (Pusede et al., 2015), and understanding its mechanisms of production is essential to managing these impacts. Surface $O_3$ is the result of both local and regional contributions when measured at any given location (Berlin et al., 2013). These contributions change in space and time because of dynamic factors that include emissions of $O_3$ precursors and meteorology. Understanding these contributions is fundamental to the design of more efficient controls on anthropogenic $O_3$ precursors to protect people and ecosystems, and to achieve compliance with the National Ambient Air Quality Standards (NAAQS) for $O_3$.

Regional contributions, often denoted as "regional background" (Berlin et al., 2013; Cooper et al., 2012), are more challenging to estimate because of variable influences from regional photochemistry and synoptic air circulation. In contrast, local contributions (e.g., from urban activities) are simply the difference between the total measured value and regional background. In the Houston-Galveston-Brazoria (HGB) area regional background $O_3$ is not well quantified on the decadal scale, likely due to lack of information on the spatio-temporal covariance of $O_3$, precursors and meteorology. Consequently, previous investigations of regional background $O_3$ in the HGB region were limited by the use of a single variable, the daily maximum 8-h average (MDA8) $O_3$ (Berlin et al., 2013). No long-term study exists that quantifies the regional contributions to direct $O_3$ precursors themselves, such as nitrogen oxides ($NO_x$ = nitrogen dioxide ($NO_2$) + nitric oxide (NO)). Our goal is to better characterize the trends in regional background $O_3$ and $NO_x$ in the HGB region on the decadal scale.

Volatile organic compounds (VOCs) also are important $O_3$ precursors. VOCs perturb the photochemical $NO_x$ cycle, the governing mechanism of tropospheric $O_3$ formation, so that $O_3$ mixing ratio increases in their presence. The relative abundance of $NO_x$ and VOCs mediates $O_3$ production through their individual reactions with the hydroxyl radical (OH). The products of VOCs' reaction with OH (peroxy radicals) react more rapidly with NO compared to $O_3$, increasing the minimum $O_3$ maintained by the $NO_x$ cycle. Therefore, VOC influence is included implicitly in the measured $O_3$ and $NO_x$ mixing ratios. In this work, we focus on the $O_3$-$NO_x$-meteorology relationship to constrain regional background $O_3$ and $NO_x$ and quantify their trends.

Meteorology influences both transport of pollutants and their chemistry. The relevant meteorological variables (wind speed (WS) and direction (WD), temperature (T), boundary layer height, etc.) and air pollution co-vary synoptically on time scales of days to weeks (Fiore et al., 2015). The effects of meteorology on tropospheric $O_3$ vary across the United States (US). Boundary layer height strongly and positively correlates with tropospheric $O_3$ in the western US (Reddy and Pfister, 2015). The $O_3$-T relationship is positive in the eastern US but weakens and turns negative along a north-south gradient, compared to the western US (Camalier et al., 2007; Tawfik and Steiner, 2013; Rasmussen et al., 2012; Reddy and Pfister, 2015). Wind speed negatively correlates with $O_3$ (Camalier et al., 2007; Banta et al., 2011; Reddy and Pfister, 2015). Wind direction can either enhance or diminish $O_3$, depending on altitude and topography-induced air circulation (Reddy and Pfister, 2015). More localized controls on decreasing surface $O_3$ include relative humidity in the southeast US (Tawfik and Steiner, 2013), shallow and deep convection in the Houston area (Langford et al., 2010a), and the intensification of southerly flow in the

HGB region (Liu et al., 2015). Recently, Wang et al. (2016) reported that the location and strength of the Bermuda High (a large scale circulation pattern) together drive the interannual variation of the monthly mean MDA8 $O_3$ in the HGB region and may either increase or decrease daily MDA8 $O_3$ during summer. Meteorological controls on the scale of the US also may play a role in the differential decline during recent decades of summer surface $O_3$ observed in the east, southeast and

midwest (Cooper et al., 2012; Hudman et al., 2009) compared to the west (Cooper et al., 2012). There are different meteorological controls in the west (i.e., thermal inversion and orographic lifting (Langford et al., 2010b)), which can either increase $O_3$ locally or transport $O_3$ up in the free troposphere and towards the east. Additionally, the pollution transport from Asia contributes to a higher $O_3$ in the western US compared to the eastern US (Cooper et al., 2012).

Synoptic air circulation contributes to ground-level $O_3$ in the HGB area in various ways. This region is influenced by the

development of high pressure centers at various altitudes during summer. Analyses of local and high altitude winds identified several such centers around the HGB region, which dictate the predominant WD (compass directions such as SW, S, SE, E, NE and N refer to the direction from which the wind originates at a given location) (Nielsen-Gammon et al., 2005; Rappenglück et al., 2008). Direct tropical storm influences from low pressure zones also were identified in the Houston area (Rappenglück et al., 2008). Dry continental air (higher $O_3$) is advected by northerly flow, industrial emissions from the Ship

Channel and Galveston Bay area are transported by easterly flow, and marine air (lower $O_3$) enters via southerly flow (Rappenglück et al., 2008). The land-sea breeze effect complicates this picture through recirculation of local pollution and formation above the coast of the Gulf of Mexico (GOM) of stagnant air masses that entrain local precursors and favor local chemistry and formation of $O_3$ (Banta et al., 2005; Darby et al., 2005; Nielsen-Gammon et al., 2005; Rappenglück et al., 2008; Langford et al., 2009).

Two intensive air quality campaigns investigated peak $O_3$ in the HGB region during 2000 and 2006, respectively (Ryerson et al., 2003; Daum et al., 2004; Banta et al., 2005; Rappenglück et al., 2008; Parrish et al., 2009; Pierce et al., 2009; Langford et al., 2010a). The $O_3$ pollution in this region was likely a result of abundant precursors emitted locally from urban and industrial sources (particularly, the highly reactive VOCs (HRVOCS) from the petroleum refineries) and the local chemistry sustained by the high summer temperature and land-sea breeze effects. However, the emissions of HRVOCs have been

considerably reduced after the first campaign, resulting in lower local contributions to $O_3$. Texas state controls on $O_3$ precursor emissions were implemented in 2007, resulting in apparent decreases in summer $O_3$ levels in the Houston area relative to the previous 8-h average NAAQS of 75 ppb (Berlin et al., 2013). It is not clear if a decline in regional background $O_3$ also contributed (Berlin et al., 2013).

Regional background $O_3$ in the HGB region has been quantified by many studies but results vary, depending on the temporal

scale, spatial scale and the altitude of observations used in data analysis (Banta et al., 2005; Darby et al., 2005; Nielsen-Gammon 2005; Rappenglück et al., 2008; Kemball-Cook et al., 2009; Langford et al., 2009; Zhang et al., 2011; Banta et al., 2011; Berlin et al., 2013; Liu et al, 2015; Souri et al., 2016). Most of the above studies used the MDA8 $O_3$ to quantify background $O_3$. Overall, regional (continental) background $O_3$ ranges from 16 to 107 ppb, while marine background has

values between 18 and 40 ppb. Local $O_3$ contributions are quantified between 25 and 80 ppb. Observations from 1-h average $O_3$ data and using wind patterns resulted in higher $O_3$ mixing ratios, particularly during stagnation in the afternoon (>140 ppb) (Darby et al., 2005). Meteorological variables, such as wind patterns, were used separately to characterize the transport regime and its diurnal transition in the HGB region and interpret their findings from data analysis; their covariance with $O_3$

and $NO_x$ was not considered.

The temporal trend in regional background $O_3$ also is still uncertain. Previous efforts to quantify the temporal trends in regional background $O_3$ from decadal surface measurements of MDA8 $O_3$ in the HGB region were made by Berlin et al. (2013). This study focused on the high $O_3$ season (May-Oct) from 1998 to 2012 and used two methods to extract the regional background $O_3$: principal component analysis (PCA) and the Texas Commission on Environmental Quality (TCEQ) method.

The former is a multivariate statistical analysis through which Berlin et al. (2013) co-varied MDA8 $O_3$ in time and space. The latter is a method used by the TCEQ and consists of manually selecting the lowest MDA8 $O_3$ measured at what are considered "background" sites (usually upwind). Using linear regression of regional background $O_3$ vs. time, Berlin et al. (2013) estimated the temporal trends and compared them to different wind quadrants. Regional background $O_3$ associated with NW winds increased over time, while that associated with SW winds remained constant. The only declining trends were

associated with the NE and SE winds, but the quantified slopes of both linear trends were highly uncertain (>50% error), suggesting that more work is needed to improve estimates of regional background $O_3$ trends. A very recent study (Souri et al., 2016) reported long-term linear trends in surface MDA8 $O_3$, which were interpreted with the help of 900 hPa wind clusters. Hence, the annual trend in regional background associated with continental air (from E-NE and E-SE) shows that MDA8 $O_3$ has declined, while that associated with marine air (from S-SE) has increased slightly, although the latter shows a

highly uncertain slope. When flow was from E-NE, it was suggested that local contributions played an equal role in declining MDA8 $O_3$. The study also did not consider covariance of MDA8 $O_3$ with meteorology and chemistry.

Regional background $NO_x$ also contributes to both surface $O_3$ and $NO_x$ in the HGB region. Through photochemistry, $NO_x$ can influence $O_3$ during transport, but it is unclear whether it enhances or diminishes the $O_3$ peaks observed locally during spring and summer. A previous study modelled both local and regional $NO_x$ summertime contributions to surface $O_3$ in

southeast Texas and found that both northern (suburban) and southeastern (coastal) sites were influenced by upwind sources (Zhang et al., 2011). The study concluded that regional $NO_x$ contributes significantly to local $O_3$ (up to 50%) and recommended regional controls on $NO_x$ emissions in addition to local controls. However, their findings are limited to 10 days and do not fully represent the seasonal and annual variations in regional $NO_x$, $O_3$, and meteorology, suggesting that a longer-term approach would refine the estimates of regional $NO_x$ contributions in the HGB region.

In this work, we estimate regional background $O_3$ and $NO_x$ by spatially and temporally co-varying chemistry and meteorology using up to seventeen years of hourly measurements and the PCA method for 8-h levels (MDA8 $O_3$ and 8-h average $NO_x$). In addition, we use two independent PCAs on $O_3$ and $NO_x$ to separately estimate regional backgrounds and test for their interaction at both 1-h (i.e., hourly median) and 8-h levels. By comparing all approaches over a period of six months, we could highlight the effect of co-varying $O_3$ with precursor and meteorology, and the effect of varying the spatial

and temporal scales. Using approaches based on continuous variables only, we quantify the temporal trends in regional background $O_3$ and $NO_x$. We compare the temporal trend in background $O_3$ with a previous study and report for the first time a decadal-scale trend in background $NO_x$.

## 2 Methods

### 2.1 Data collection and processing

Public data, representing 1-h average surface measurements of $O_3$, $NO_x$ and meteorology (WD, WS and T), were downloaded from the Texas Air Monitoring and Information System website owned by TCEQ (see Data availability). The measurements were taken every second, averaged over five minutes and then averaged over one hour. Note that, due to the measurement method (combined chemiluminescence detection-molybdenum conversion), the monitored total $NO_x$ might include traces of other oxidation products (PAN, $HNO_3$, etc.). The locations of the monitoring sites are mapped in Fig. S1 (in Supplemental Information). For each site, we generated and exported raw data reports (validated data only) for the period of May-Oct. 1998-2014. Using the hourly measurements, we computed three variables to be used in the estimation of background $O_3$ and $NO_x$: the hourly median per month, MDA8 $O_3$, and 8-h average $NO_x$ corresponding to MDA8 $O_3$.

The hourly median was used for two purposes: (1) replacement of missing values, ensuring that multiple parameters are available at the 1-h level for multivariate data analysis, and (2) use in the analysis as a variable itself because it is a highly representative value, derived from many replicates of each daytime hour (i.e., years of observations) at various sites. Overall, up to 5% of the missing raw data was replaced by the hourly median (Fig. S2). The protocol for filling data gaps was to replace no more than six consecutive hours in a day (i.e., 25% of the day missing). Therefore, gaps from one to six hours were identified and replaced with the corresponding hourly median. Ten sites have data coverage for 13 years, and five sites have the largest data coverage for 17 years. Therefore it was possible to observe changes in background $O_3$ and $NO_x$ over a time-scale of almost two decades, but the spatial coverage was limited to just five sites. Berlin et al. (2013) also identified six nearly continuous sites (five identical to those identified in this study) using directly the MDA8 $O_3$ from the same data source (not hourly data as we used here to calculate the MDA8 $O_3$). However, in our study, a one-decade analysis was also possible by doubling the number of sites, thus increasing slightly the spatial scale for analysis.

We ran a preliminary bi-variate site correlation analysis from five sites within the HGB area and found that the time-scale of variability in $NO_x$ is much smaller than that of variability in $O_3$, affecting the correlation of hourly median $NO_x$ between sites. Therefore, $NO_x$ appears to be more sensitive than $O_3$ to fast changes in meteorology, for example. The temporal scale of analysis should be relevant to both $O_3$ and $NO_x$ variabilities in order to test if there is any chemical interaction between them during transport, which could influence the estimation of background levels. An hourly median approach, in combination with those focused on 8-h averages, would allow for observation of the effect of temporal scale in the monthly trends of background $O_3$ and $NO_x$.

## 2.2 Data analysis

We used PCA to analyze single and multiple variables at various sites in the HGB area. The PCA method is a data reduction technique that uses the framework of linear algebra (eigenvector and eigenvalues) to reduce a larger data set to a smaller one, based on common modes of variance or strong correlations among variables (Wilks, 1995). In PCA, a non-square matrix n x K (i.e., time x space or site value) is converted to a square matrix K x K (variance-covariance or correlation matrix). The off-diagonal elements of the correlation matrix are important as they reflect the correlations of one or more variables at each location to any other location, while the diagonal elements are 1, representing the autocorrelation of each site in terms of the variable considered. This correlation matrix is transposed to compute an eigenvector matrix (or component matrix) of which elements are the loadings or the Pearson's correlation coefficients, if the correlation matrix is used instead of the variance matrix. The loadings range from -1 to +1 (the highest correlations possible) with a mean of 0 (no correlation). By summing the squared loadings of each component (column) we obtain the eigenvalue of that component. By squaring the loadings and summing them from all components for each variable (row), we get the maximum variance that could be explained by all the components, which is 1. This is not always the case, as not all the components are retained. For example, the maximum number of the components that can result from PCA equals the number of the original variables. In general, the first few components explain most of the variance in the original variables, while the remaining components explain very little. If only the first components are retained, then their squared loadings must be normalized by their respective sum (which is less than 1). These normalized values can be used to convert the PC scores (standardized regression coefficients) to original variables (Wilks, 1995; Langford et al., 2009). The PC scores (also negative and positive) are the elements of the new variables (components) and they have a wider range than the loadings. The resulting PCs are unique and distinct due to the eigenvectors being perpendicular to each other. However, the fact that PCs are orthogonal and distinct is not enough to account for their physical meaning. Therefore, PCA uses rotation techniques (i.e., Varimax) to rotate the eigenvectors; thus, in addition to the fact that they are distinct from each other, they also have a physical meaning based on the association of the significant elements they contain (i.e., loadings). The output of this rotation is the rotated component matrix which has a different composition of loadings than the unrotated one. The percentage of the variance explained by each component also changes. We rotated the components in this study. Using the PCA method implemented in the IBM SPSS Statistics 24 software, we used different approaches to extract regional background of $O_3$ and $NO_x$ from locally measured values that were converted to hourly median, MDA8 $O_3$ and 8-h average $NO_x$ for analysis, as described below. In addition to PCA, we used linear regression of season-scale background $O_3$ and $NO_x$ vs. time (year) to quantify temporal trends. We also used linear regression to test for chemical interaction, to quantify how much the change in regional background $O_3$ could be explained by the change in regional background $NO_x$, and to estimate the regional contributions to locally observed $O_3$ and $NO_x$.

### 2.2.1 PCA of hourly median to estimate regional background $O_3$ and $NO_x$ and other contributions

To estimate the characteristic hourly regional background $O_3$ and $NO_x$, we used the hourly median described in Sect. 2.1 for 28 monitoring sites (Table 1 and Fig. S1) when it could be determined from the available measurements during 1998-2014. Two independent PCAs of median $O_3$ and $NO_x$ were run using daytime hours (local 10 am - 6 pm), over a period from May to October (eight median values for each month). In this approach, new from the perspective of the metric used in the PCA, we did not co-vary $O_3$, $NO_x$ and meteorology as their respective hourly medians may not always represent coincident measurements of all of them. Instead, we used meteorology to interpret the PCA results as previous studies did.

### 2.2.2 PCA of MDA8 $O_3$ and 8-h average $NO_x$ to estimate regional background $O_3$ and $NO_x$ (Approach A)

In this approach, we used two independent PCAs on daily MDA8 $O_3$ and the corresponding 8-h average $NO_x$ to extract the regional backgrounds, but fewer sites were used than in the hourly median approach (5 vs. 28). Here we only considered sites with quasi-continuous data for the longest period possible (17 years) to estimate more accurately the regional background. These sites are all within Harris County: Aldine, Bayland Park, Deer Park, Houston East and NW Harris (Fig. S1). Like in the previous approach, we only used meteorology to interpret the principal components.

The MDA8 $O_3$ was used in previous studies to estimate background $O_3$ (Nielsen-Gammon et al., 2005; Langford et al., 2009; Berlin et al., 2013; Souri et al., 2016), but no study looked at background $NO_x$ using coincident measurements from the same sites. To compare temporal trends obtained from this study with other studies (Berlin et al., 2013; Souri et al., 2016), we separately ran PCA for $O_3$ and $NO_x$. Additionally, we compared the background estimates from this approach with those obtained from the hourly median approach to isolate the effect of time-scale (which influences the dynamics of the 6-month trends) and with other approaches in this study (subsequent sections) to isolate the effect of chemical and meteorological interaction within the HGB area.

### 2.2.3 PCA of MDA8 $O_3$ and 8-h average $NO_x$ to estimate regional background $O_3$ and $NO_x$ (Approach B)

As a novel approach, we ran five multivariate PCAs for each site (the same sites and period used in the previous approach) to constrain the estimation of background $O_3$ in the HGB area with chemistry and meteorology and to improve the quantification of its temporal trend. This approach is different from those described in previous sections and studies (single variable, multiple sites) because it takes into account more variables (multiple variables, single site). The variables considered at each site are MDA8 $O_3$ and the corresponding 8-h average $NO_x$, WD, WS, and T.

### 2.2.4 PCA of MDA8 $O_3$ and 8-h average $NO_x$ to estimate regional background $O_3$ and $NO_x$ (Approach C)

This approach is similar to Approach B except that we used more sites (10) and a shorter period of time (13 years), based on simultaneous data availability and continuity at these sites. The five additional sites are: Clinton, Channelview, Manvel Croix, Seabrook Friendship Park and Conroe Relocated (Fig. S1). Use of larger spatial data coverage could improve the

estimation of regional background, even if the study period is shorter, because it would capture variations in chemistry and meteorology within the HGB area.

## 3 Results and Discussion

### 3.1 Hourly median approach

#### 3.1.1 Main regional contributions to hourly median $O_3$ and $NO_x$

The PCA resulted in four components for $O_3$ and five components for $NO_x$. Only components with eigenvalues greater than 1 were retained. The first components explained most of the percentage of the variance in original $O_3$ and $NO_x$ (~51% and ~45%, respectively) and were highly correlated at more than half of the initial sites (16 out of 28). Among these "PC1 sites," 12 are common sites for both $O_3$ and $NO_x$.

An interesting cluster-like pattern emerged when we mapped the sites that highly correlated with any of the PCs (e.g., loadings with absolute values of 0.5 or higher). The sites associated with these loadings (Table 1) are mapped in Fig.1, in which different point sizes are used to show the overlapping of both $O_3$ and $NO_x$ sites, while color is used to show the correlation of the same component at various sites (i.e., clusters). The widespread cluster (PC1) suggests a larger-scale control on both $O_3$ and $NO_x$, while the smaller cluster (PC2) suggests a more localized control. The proximity to the GOM

emphasizes that PC1 is largely influenced by marine background during summer. The proximity to the Houston Ship Channel indicates that PC2 likely represents local effects (i.e., chemistry, emissions, etc.). Given the proximity to the rural area in the north of the HGB region, PC3 might represent a mix between regional (continental) and local (urban) contributions.

The spatial patterns of the components, their extents and locations within the HGB region all indicate that PC1 represents

regional background for both $O_3$ and $NO_x$. We arrive to this finding by spatially interpolating the three main clusters from Fig. 1 to reveal continuous patterns of correlations (Fig. 2). The $O_3$ pattern for the first component (the square-like pattern in the south of the HGB region) emphasizes the marine influence because of the higher loadings along the coast, while the lowest loadings are within the region overlapping with the second component, where local effects seem to be more important (the smaller rectangle in the proximity of the Houston Ship Channel). The PC1-derived $NO_x$ pattern shows high correlations

in the same area pointed out by PC1-$O_3$, but the highest correlations appear in the west of the Bay area; lower loadings also occur in the area controlled by the local effects.

Meteorology also supports the hypothesis that PC1 describes regional contributions and reveals that these are mostly marine in summer and continental in spring and fall. To test if PC1 is regional background, we plotted the PC1-$O_3$ and PC1-$NO_x$ scores against WD and WS in Fig. S3a-e. Overall, two flow regimes explained the changes in PC1-$O_3$ (Fig. S3a): summer

(marine) flow decreases PC1-$O_3$ (negative scores), while spring/fall (continental) amplifies it (positive scores). There was no sign of stagnation in summer (an increase in PC scores at lower WS) from which we could infer local chemistry (Fig. S3b).

The PC1-NO$_x$ tells roughly a similar story in terms of flow regimes (Fig. S3d) and the absence of stagnation during summer (Fig. S3e). Temperature indicates no consistent formation of O$_3$ with increasing T at the scale of the entire season (although the monthly relationship is positive) and very limited chemistry or some physical effect on NO$_x$, such as dilution at the surface due to a higher boundary layer (Fig. S3c and Fig. S3f).

The monthly background O$_3$ and NO$_x$ trends are consistent between hours over the entire season. We determined this by converting the PC1 scores to O$_3$ and NO$_x$ hourly mixing ratios and plotting them for each month to assess the 6-month trends (Fig. S4). Background O$_3$ trends compare well with those from previous estimates of 8-h average background O$_3$ (Nielsen-Gammon et al., 2005), showing two peaks in spring and summer/fall, respectively, and a drop in mid-summer, when local chemistry dominates regional background O$_3$ in the HGB region.

The season-characteristic hourly background O$_3$ and NO$_x$ (the most typical daytime value on 1-h basis in the HGB region averaged over six months) points out consistency between hours and no significant chemistry between O$_3$ and NO$_x$ (Fig. 3), particularly during midday, when important photochemistry occurs. When the 6-month values are also averaged over 8 hours, they compare reasonably well with similar estimates from previous studies (Nielsen-Gammon et al., 2005; Choi, 2014), ranging from 37 to 38 ppb for background O$_3$ and varying between 4-7 ppb for background NO$_x$.

We further assessed the relationship between regional background O$_3$ and NO$_x$ at both 1-h and 8-h levels (Fig. S5). The positive relationships suggest that both O$_3$ and NO$_x$ are related (possibly through regional transport) and there is some interaction between them (significant slopes of 1.89 ± 0.48 and 2.07 ± 1.99, respectively). However, background NO$_x$ only explains ~60% of the changes in background O$_3$, at both 1-h and 8-h levels, implying that the unexplained ~40% might be related to other processes/sources, such as regional VOC chemistry or from unconsidered VOC emissions upwind, which can increase both O$_3$ and NO$_x$ mixing ratios. It is also possible that a fraction of background NO$_x$ (including lightning NO$_x$) was converted to PAN and HNO$_3$, which was accounted for in the total NO$_x$ by the measurement method, reducing the potential of background NO$_x$ to explain background O$_3$. Stratospheric O$_3$ also may explain some of the background O$_3$ in the HGB. However, stratospheric O$_3$ contributions are either overestimated at mid-latitudes by the global cross-tropopause transport models (Liu et al., 2016) or the relationship between the cosmogenic beryllium-7 associated with particulate matter and surface O$_3$ observed in the HGB region is not conclusive enough (Gaffney et al., 2005). Modelling based estimates of lightning NO$_x$ in the GOM suggest that this source is negligible near the surface, ranging from near zero to 50 ppt during two summer months (Pickering et al. 2016).

**3.1.2 Other contributions to hourly median O$_3$ and NO$_x$**

Here, we report results from the analysis and interpretation of the other significant components (PC2-PC5) extracted by PCA

30    using the hourly median approach. The cluster of points localized around the Houston Ship Channel, where most of the petrochemical industry facilities are located, is likely related to local chemistry and/or emissions. The cluster of points representing highly correlated PC3 with both O$_3$ and NO$_x$ at locations in the north of the HGB area (Fig. 1) likely represents a mixed local/regional (maybe continental) influence. Additionally, it was important to consider how the other components

(PC4 for $O_3$ and PC4 and PC5 for $NO_x$) may factor into the average of local contributions within the HGB region, since the sites defining them are in close proximity to the PC2 sites, from which we primarily inferred local contributions.

The second component describes local contributions, given the locations of the sites and its relationship with meteorological variables. To test for local influence, we analyzed the PC2-$O_3$ and PC2-$NO_x$ scores against meteorology (Fig. S6). Results revealed that PC2 is insensitive to WD for both $O_3$ and $NO_x$ at the season scale using 1-h and 8-h levels. Within the high $O_3$ season, flow varies from SSW-S-SSE (in summer) to SE-ESE (in spring and fall). Highest PC2-$O_3$ scores are recorded in July and August, coinciding with the predominant flow from SSE-SE. A few high scores are also visible in September, but they appear to be related to easterly transport. Overall, the spring and fall PC2-$O_3$ scores all cluster under zero at relatively similar flow direction as observed in summer. This suggests some local effects, a reverse pattern than that inferred from PC1-$O_3$ in Fig. S3a. Local effects can also be inferred from PC$_2$-$NO_x$, with highs and lows in each month (Fig. S6d). Diurnal variability in PC2-$NO_x$ scores is more pronounced for $NO_x$ compared to $O_3$, suggesting that $NO_x$ is lost photochemically in the afternoon hours (i.e., lower scores). With respect to WS, PC2-$O_3$ and PC2-$NO_x$ show different relationships (Fig. S6b and Fig. S6e). Low WS facilitates the formation of $O_3$ and depletion of $NO_x$. As WS increases (> 4 m s$^{-1}$) $NO_x$ increases (higher PC2-$NO_x$ scores) but there is no sign of $O_3$ formation (low PC2-$O_3$ scores).

Relationships with temperature suggest active local chemistry by both month and season (Fig S6c and Fig. S6f). A positive PC2-$O_3$ versus T relationship indicates the build-up of $O_3$ as temperature increases to favor the chemistry of VOCs. A negative PC2-$NO_x$ versus T relationship may suggest both chemical and physical controls on $NO_x$. However, the high scores in July and August might be related to $NO_x$ and VOCs chemistry, rather than vertical mixing due to a higher boundary layer. Therefore, we interpreted that PC2 represents mainly local chemistry. To test if PCA-inferred local $O_3$ is explained by PCA-inferred local $NO_x$, the converted PC2 variables are compared in Fig. S7. The negative relationship is consistent with $NO_x$ chemistry and photochemical production of $O_3$; it also indicates the probability of a VOC-limited atmosphere. However, $NO_x$ only explains about 30% of the changes in $O_3$. Note that the 8-h average did not reveal a significant dependence of $O_3$ on $NO_x$ at the season scale (the empty circles), pointing out the importance of the time scale (1 h) needed to observe relevant chemistry. The unexplained portion for the 1-h level (70%) is quite significant. We believe it is related to rapid VOC chemistry in this area of the HGB region. Daum et al. (2004) measured various plumes for almost two weeks in late summer of 2000 and showed that six of them were different from typical urban plumes: they were rich in formaldehyde and peroxides, attributable to hydrocarbon oxidation and photochemistry, respectively. They also found that $O_3$ formation in these plumes was very efficient (6.4-11 ppbv $O_3$/ppbv of $NO_x$). These plumes were tracked back to sources of $NO_x$ and hydrocarbons in the proximity of the Houston Ship Channel. Using zero-dimensional model predictions, they found that $O_3$ formed very fast (140 ppbv/h). Compared to urban plumes, the authors found that the formation of $O_3$ in plumes from the Ship Cannel was more $NO_x$-limited, but uncertainties remain whether the production of $O_3$ in this area is $NO_x$- or VOC-limited.

The third component may be dominated by regional influences, based on the locations of the associated sites within the HGB region and the comparison with meteorology. Traditionally, the upwind sites (Conroe, Conroe Relocated, NW Harris) are

considered to be "background" sites. One PC3 site (Houston Aldine), though, overlaps with a PC2 site resulting in a mixed contribution within PC3 at this site (Fig. 1). To consider mixed regional/local influences, the PC3-$O_3$ and PC3-$NO_x$ scores were examined with respect to meteorological variables. In the morning, flow is from the GOM, which brings already processed air, characterized by low PC-$O_3$ scores (marine background); PC3-$NO_x$ scores vary from positive to negative

within this onshore flow. In the afternoon, flow is from the SSE-SE and intercepts some local/urban pollution on its way to the PC3 sites (i.e., Conroe); here, PC3-$O_3$ increases (continental background), while PC3-$NO_x$ varies largely. Temperature increases PC3-$O_3$ while decreasing PC3-$NO_x$, suggesting active chemistry by both month and season. Winds are stable and stagnant in the afternoon, suggesting enhanced local pollution during that time. At the season scale, the $O_3$-WS relationship is positive, while the $NO_x$-WS relationship is positive during spring and summer months only, turning negative in fall. The

positive relationship suggests advection of higher mixing ratios of both $O_3$ and $NO_x$ to the HGB area, while a negative relationship suggests a chemical or a physical loss of $NO_x$. The former indicates that regional contributions may dominate the local contributions within this component at the season scale (for $O_3$) and during spring and summer (for $NO_x$). Covariance with meteorology would probably better resolve PC3, but this approach was not possible using the hourly median.

The fourth component likely describes local transport effects. Results from analysis of PC4-$O_3$ and PC4-$NO_x$ while considering meteorology indicate that the sites associated with this component (Clinton, La Porte) are influenced by the sea breeze rotation and recirculation of local pollution (flow is from S-SSE in summer/spring and from SE-ESE in fall), with higher scores occurring in spring/summer.

The fifth component, which explained a small portion of the variance in original $NO_x$, appears to be consistent with local

VOC chemistry because its relationship with T is positive over the entire season. Primarily, $NO_x$ increases in summer due to VOC chemistry and/or local emissions. On a monthly basis, PC5-$NO_x$ is negative with increasing T (similar to PC2-$NO_x$), suggesting physico-chemical controls on $NO_x$. Flow is from SSE-SE-ESE and winds are weak and stable (~3 m s$^{-1}$) in summer (increases $NO_x$) and less stable in spring/fall (decreases $NO_x$). On a monthly basis, PC2-$NO_x$ and PC5-$NO_x$ are not very different, as they both may be controlled by physico-chemical interactions involving boundary layer height, solar

radiation, VOC chemistry and possibly other chemistry. However, if we extend the time scale to six months, the two components are very different in terms of the $NO_x$-T relationship: PC2 is negative, while PC5 is positive with increasing T. A possible explanation is that the two components, when compared to T, are different because of the averaging over 8 hours. These averages are consistent with the 1-h based PC2-T relationships, but are inconsistent with the 1-h based PC5-T relationships. Consequently, the $NO_x$-T relationship turns positive for PC5 at the season scale. On the other hand, in this

PCA approach, we did not use 8-h averages and T, but the method differentiated between PC2 and PC5. A possible explanation is that one of the PC5 sites (Baytown) overlaps with the PC2-defined cluster in Fig. 1, being more exposed to local chemistry and emissions from the industrial area, an influence standing out at the season scale only. La Porte is situated south of the Houston Ship Channel and near the GOM, likely being dominated by marine influences (lower $NO_x$) at the monthly level. Therefore, PC5 also describes mixed local/regional effects on surface $NO_x$.

We primarily based our regional background $O_3$ and $NO_x$ estimates on PC1, although some regional contributions could be inferred from other components (most notably, PC3). Since the components from which we inferred mixed regional-local contributions explain less variance than PC1 (particularly, PC5), we assumed these contributions are negligible, so we did not include them in the estimation of regional background $O_3$ and $NO_x$. Similarly, we estimated local $O_3$ and $NO_x$ from the conversion of PC2 only. However, for estimating the contribution of regional background to measured hourly median $O_3$ and $NO_x$, we additionally considered average regional contributions from PC1 and PC3 and compared them with those estimated from PC1 only.

## 3.2 Regional and local contributions to MDA8 $O_3$ and 8-h average $NO_x$ (Approach A)

The two independent PCAs using fewer sites with nearly continuous data for which the MDA8 $O_3$ and 8-h average $NO_x$ could be calculated resulted in three components having eigenvalues greater than unity. However, we retained all five components because they were not significantly different in explaining the variance in the original variables, particularly for $NO_x$; their loadings are shown in Table 2. PC5 was not significant for $O_3$.

Meteorology helped to interpret the components but was insufficient to clearly distinguish between regional and local contributions. For example, by looking at how the scores of each component varied with average WD we found that all sites were influenced by SSE winds (146-155 degrees), with the western sites (NW Harris and Bayland Park) experiencing a slightly more southern WD by 3 degrees. The flow from GOM encounters local/urban air on its way to the western sites, while eastern sites experience more direct marine air from the GOM area. These two patterns were also visible in the distributions of PC scores vs. average T and WS.

Monthly trends helped to distinguish between regional and local contributions from the principal components. We used the monthly trends for each component to observe if these trends are consistent with expected regional and local trends from previous studies. Three components (PC2, PC3 and PC4) exhibit monthly trends (Fig. S8a) that are consistent with the expected bi-modal regional background $O_3$ (Nielsen-Gammon et al., 2005). The remaining components (PC1 and PC5) show monthly trends (Fig. S8b) similar to those expected from unimodal local contribution (Nielsen-Gammon et al., 2005). We found similar monthly trends for 8-h average $NO_x$ (Fig. S8b). Here, regional contributions are suggested by PC1, PC2 and PC5, while local contributions are denoted by PC3 and PC4. Therefore, we based our regional and local estimates of $O_3$ and $NO_x$ on the components identified as regional and local from their monthly trends.

The relationship between regional background $O_3$ and $NO_x$ (Fig. S9) underscores that $NO_x$ explained approximately 20% of the changes in background $O_3$, while no significant relationship between PCA-inferred local $O_3$ and $NO_x$ was observed (Fig. S10). These poor relationships may be the result of using fewer sites, MDA8 $O_3$, and 8-h average $NO_x$ compared to the hourly median approach.

**3.3 Regional and local contributions to MDA8 $O_3$ and 8-h average $NO_x$ (Approach B)**

In this new PCA approach, we co-varied $O_3$ with $NO_x$ and meteorology at the sites used in Approach A. We conditioned the PCA to retain only components with eigenvalues greater than 1. Two components were retained at each site. The average eigenvalue was 1.5. Each component explained approximately 30% of the variance in the original variables, implying that they are equally important in explaining the original variables at the sites used in this approach.

We partially inferred the meaning of the components by considering how variables and their respective loadings (absolute values nearly or greater than 0.5) are associated within each component (Table 3). The first component (PC1) associated $O_3$ with WS and, sometimes, with $NO_x$ at three sites (Bayland Park, Deer Park and NW Harris), while the same component combined $NO_x$ with T at other sites (Houston Aldine and Houston East). On the other hand, the second component (PC2) associated $O_3$ with WS at two sites (Houston Aldine and Houston East) and combined $NO_x$ with T at the remaining sites (Bayland Park, Deer Park and NW Harris). Overall, two patterns emerged from each component: "$O_3$-$NO_x$-WS" sites and "$NO_x$-T" sites. The association of $O_3$ with WS could indicate a physical control (i.e., advection or stagnation), while the $NO_x$-T relationship may suggest a chemical control (T-mediated chemical reactions). In the first component, $O_3$ and WS also associate with $NO_x$ (with lower loadings), suggesting either some chemical interaction sustained by a lower WS or a similar transport source for both $O_3$ and $NO_x$. Temperature and $NO_x$ at Houston Aldine confirmed that "$NO_x$-T" in the first component describes chemistry, possibly local formation of $O_3$ (Fig. S11). Ozone, $NO_x$ and WS at Bayland Park together confirmed that "$O_3$-$NO_x$-WS" represents regional transport of $O_3$ and $NO_x$ and/or local VOC chemistry, because both $O_3$ and $NO_x$ increase with PC1, while WS decreases (Fig. S12). Local chemistry might be possible at lower WS, which causes an increase in PC1 scores.

By mapping how the input variables are partitioned between the two components we more clearly discriminated between regional and local contributions at each site (Fig. S13). For instance, $O_3$ is well represented by PC1 at three sites (NW Harris, Bayland Park and Deer Park). At these sites, some $NO_x$ is also distributed in PC1, suggesting that $O_3$ and $NO_x$ are related either through transport or chemistry. However, WS shows a pattern strongly similar to that of $O_3$ and less strongly to that of $NO_x$ in PC1, reinforcing that PC1 at these sites is dominated by regional transport. At Houston Aldine and Houston East, $O_3$ shows an opposite partition compared to $NO_x$, indicating that PC1 at these sites is local chemistry, which also is supported by T and WS.

Regional background $O_3$ and $NO_x$ were determined by averaging the converted PC scores from "$O_3$-$NO_x$-WS" sites, while local contributions were quantified by averaging the converted PC scores from "$NO_x$-T" sites. The conversion method (Langford et al., 2009) differs slightly from Approach A because in Approach B multiple variables defined one component at a particular site as opposed to a single variable at many sites. Therefore, the normalized relative contribution (in %) of the variable of interest in each component was used instead of the total variance (in %) explained by the component.

**3.4 Regional and local contributions to MDA8 $O_3$ and 8-h average $NO_x$ (Approach C)**

The simultaneous effect of increasing the spatial scale and reducing the temporal scale of the analysis (constrained by the availability of continuous data) was studied using Approach C. Therefore, results in this section were driven by the use of five more sites and a shorter study period compared to Approach B. The same variables were used in PCA as in Approach B. For each site, there were two components retained (average eigenvalues of 1.3-1.6) and each explained, on average, 31% and 27 % of the variance in MDA8 $O_3$ and 8-h average $NO_x$, respectively. Similar to Approach B, we also identified two modes of variance among the original data: "$O_3$-$NO_x$-WS" (denoting a physical control) and "$NO_x$-T" (denoting a chemical control) based on loadings in Table 4 (those with absolute values nearly or greater than 0.5). Therefore, we obtained the regional background $O_3$ and $NO_x$ by averaging the corresponding PC scores and using the adjusted equation from Langford et al. (2009) as described previously.

**3.5 Similarities and differences between monthly trends of regional background $O_3$ and $NO_x$ from all approaches**

We compared the monthly trends from all approaches used to estimate regional $O_3$ and $NO_x$ contributions. We found that the use of MDA8 $O_3$ (Approaches A-C) estimated larger background contributions for the entire season compared to the hourly median approach (either from PC1 only or from PC1 adjusted by PC3), as shown in Fig. 4. This likely is due not only to the difference in the number of sites used in the PCA (5-10 vs. 28, respectively) but also to the fact that the highest 8-h average was selected for each day in Approaches A-C, compared to the hourly median (the $50^{th}$ percentile of the hourly measurements), which was averaged over 8-h for comparison. In Fig. 4, the hourly median approach also reveals a stronger onshore effect than the MDA8 $O_3$ approach. This could be because of the smaller time scale of observations, which allows the median to capture better the influence of the onshore flow in terms of $O_3$. Approach A follows the trend described by the hourly median (although smoothed) because it was derived using a similar PCA (single variable/multiple sites). Approaches B and C deviate from this trend because they were derived using a different PCA (single site/multiple variables). Regardless of the approach, background $O_3$ drops in July, which is consistent with the bimodal variation of the annual 8-h average background $O_3$ (Nielsen-Gammon et al., 2005) and with the less intense and a more easterly Bermuda High during July (Wang et al., 2016). The three approaches (A-C) yield similar values for July, when local chemistry is expected to be more important (Nielsen-Gammon et al., 2005). The sudden increase from July to August is consistent in all approaches (significant regional summertime chemistry), but background $O_3$ starts decreasing earlier for Approaches B and C compared to the hourly median and Approach A, likely the result of changes in meteorology after August (less influence from sea breeze effects). Because meteorology was not used to estimate regional background $O_3$ in the hourly median approach or in Approach A, the enhancement of background $O_3$ continues until September and starts declining only after, as a result of changing regional transport and chemistry. Interestingly, approaches B and C agree with the hourly median approach in May and October, suggesting that the time scale of observations (1-h) is small enough to capture rapid changes in $NO_x$ concentration and fluctuations in WS, which are reflected in the 8-h average regional background $O_3$.

A similar analysis was done for regional background $NO_x$ (Fig. 5). Here, estimation of larger background $NO_x$ resulted from Approaches A-C until mid-August, when compared to the hourly median approach based on PC1 only. All approaches intersect this hourly median approach sometimes between August and September. However, when the regional background from the hourly median approach is adjusted by PC3 (average of PC1 and PC3), Approaches A-C all gave higher estimates than the hourly median over the entire season. Approach A appears consistent with the hourly median "adjusted by PC3", for the same reasons described previously for background $O_3$. The effect of spatial scale is more visible between Approaches B and C from August to September, when local influences likely dominate within the HGB region.

## 3.6 Quantification of temporal trends in regional background $O_3$ and $NO_x$

The goal in this portion of the work was to quantify the temporal trends in the final background $O_3$ and $NO_x$ and to investigate if the background $O_3$ and $NO_x$ have declined over the past decades. In addition, we wanted to assess the effects of co-varying chemistry and meteorology on these trends. We used linear regression of the season- averaged background $O_3$ and $NO_x$ in each year vs. time to quantify temporal trends.

### 3.6.1 Weak and negative linear trends resulted from Approach A

The temporal trend quantified from Approach A (Fig. 6) suggests that background $O_3$ has declined; corresponding average WD also is shown for the five sites. The linear model is statistically significant, yielding a slope of -0.13 ± 0.10 ppb $y^{-1}$, comparable in magnitude but smaller than that reported in a previous study and irrespective to WD (Berlin et al., 2013) using a similar approach (-0.33 ± 0.39 ppb $y^{-1}$). Compared to the SE wind-constrained slopes from Berlin et al. (2013) (-0.92 ± 0.74 ppb $y^{-1}$ or -0.79 ± 0.65 ppb $y^{-1}$), our slope is much smaller but closer to that from Souri et al. (2016) (0.09 ± 0.40 ppb $y^{-1}$). The mean background $O_3$ over the seventeen years is 46.74 ± 0.58 ppb and compares well with the 14-y and 15-y means from Berlin et al. (2913) and Souri et al. (2016) (42.5 ± 6.3 ppb and 57 ± 19 ppb, respectively), representing SE influences only. The decadal time-scale explained about 27% of the changes in background $O_3$ in this study, similar to Berlin et al. (23%).

The decline in background $NO_x$ is better explained by this approach ($R^2$=0.53) compared to $O_3$, due to less scatter in the data after 2003, while the slope is similar compared to that for $O_3$ (Fig. 7). On average, the 17-y background $NO_x$ is 6.86 ± 0.19 ppb. Note that due to potential biases in background $NO_x$ (p.15-16 in the SI), this value represents the upper bound in background $NO_x$. After taking into account the overall bias, we also estimated a lower bound in background $NO_x$ of 4.49 ± 0.12 ppb (see Table 5 for all approaches). The linear trends for all approaches were shifted to lower ranges by ca. 2 ppb, on average (Fig. S22).

### 3.6.2 The negative trend significantly improved for $O_3$ using Approach B

When background $O_3$ is adjusted by $NO_x$ and meteorology, its decline over time is stronger and more significant than in Approach A (Fig. 6), though still of the same order of magnitude. The resulting slope is -0.68 ± 0.27 ppb $y^{-1}$, while the 17-y mean of background $O_3$ is 46.72 ± 2.08 ppb, in agreement with the previous approach. Relative to a previous study (Berlin et al., 2013), the slope is less steep (-0.69 vs. -0.92 ppb $y^{-1}$ or -0.79 ppb $y^{-1}$), but its error is halved (42% vs. 80%, respectively). Our slope, though smaller, compares well in terms of absolute error with the slope from Souri et al. (2016), describing continental regional background $O_3$ (-1.0 ± 0.55 ppb $y^{-1}$); however, as Souri et al. (2016) suggested, local sources may have contributed half to the observed $O_3$ within the E-NE wind cluster, which could explain the steeper slope observed in their study. They also reported a weaker slope for regional background $O_3$ from the E-SE (-0.9 ± 0.86 ppb $y^{-1}$). As observed in Fig. 6, a slight shift in WD over the past seven years (more southerly flow) might have also played a role in the decline of background $O_3$, which is consistent with the findings in Liu et al. (2015). Also, State of Texas controls on precursor emissions implemented in 2007 (Berlin et al., 2013) may also have contributed to reduced background $O_3$ after that.

The slope of background $NO_x$ versus time is slightly smaller compared to Approach A (-0.04 ppb $y^{-1}$ vs. -0.06 ppb $y^{-1}$), but the linear model performed better ($R^2$= 0.58 versus $R^2$=0.53), highlighting the effect of spatial and temporal covariance of chemistry and meteorology (Fig. 7). The 17-y mean of background $NO_x$ (6.80 ± 0.13 ppb), representing the upper bound, is in good agreement with Approach A. The average value corresponding to the lower bound of background $NO_x$ is 4.45 ± 0.08 ppb.

### 3.6.3 The negative trends did not improve using Approach C (spatial extension of Approach B)

By extending the spatial scale (from 5 to 10 sites) and lowering the period of analysis (from 17 to 13 years), the effect of co-varying $O_3$ with $NO_x$ and meteorology within the HGB area did not make a significant difference in the temporal trend of background $O_3$ (Fig. 6), but it weakened the temporal trend in background $NO_x$ (Fig. 7). It is possible that $NO_x$ from additional sites was more sensitive to local influences (i.e., meteorology) than $O_3$ or that the years left out from analysis had higher 8-h average $NO_x$ mixing ratio. The 13-year mean of background $O_3$ is 44.71 ± 1.28 ppb, while of mean upper bound background $NO_x$ is 6.03 ± 0.05 ppb. The lower bound estimate of mean background $NO_x$ represents 3.95 ± 0.03 ppb.

### 3.7 Regional background contributions to locally measured $O_3$ and $NO_x$ from all approaches

We quantified the regional background contributions to locally measured $O_3$ and $NO_x$ via linear regression for all the approaches in this study (Fig. S14 to Fig. S21). Based on slope values, these contributions ranged from 1.16 to 5.65 (mole measured per mole of background) for measured $O_3$ (hourly median and MDA8) and varied from 0.33 to 4.06 for measured $NO_x$ (hourly median and 8-h average). Compared to the analogous slope from Berlin et al. (2013) (1.22 ± 0.04), our slope value for $O_3$ using approach A is about five times steeper (5.65 ± 0.15), while those from approaches B and C are slightly lower (0.91 ± 0.02) or slightly higher (1.47 ± 0.06), respectively. The intercept coefficients were significant in all

approaches. Background $O_3$ explained between 57 % and 98 % of the variation in spatially averaged hourly median and MDA8, while background $NO_x$ explained about 16-62 % of the changes in spatially averaged hourly median and 8-h average. In general, the linear model performed less well for $NO_x$ (all approaches) compared to $O_3$. This could be explained by its smaller temporal scale of variability compared to $O_3$ but also by the fact that the corresponding 8-h average $NO_x$ to MDA8 $O_3$ was used in the PCA. It is possible that this approach makes it more difficult to extract background $NO_x$, if MDA8 $O_3$ is mainly the result of local chemistry (see p. 15-16 in the SI for potential biases). The larger estimates of background $NO_x$ compared to measured median values from May through October could be the result of a stronger intra-seasonal variability for $NO_x$ (Fig. S15). For example, the measured median relates negatively with background $NO_x$ from May to July (the cluster around 5 ppb); it only turns positive after that, from July to October. As a consequence, hourly background $NO_x$ is overestimated in spring compared to summer and fall and relative to measured values. A separate analysis of hourly median $NO_x$ within the PCA for spring vs. summer/fall potentially could improve the estimates of the upper bound of background $NO_x$ using the hourly median approach. Also, it should be noted that background $NO_x$ was not adjusted by meteorology, as their covariance was not possible using the hourly median.

### 3.8 Summary

Approach B is our best estimate of the temporal trend in background $O_3$. Results from all approaches are summarized in Table 5, along with values from Berlin et al. (2013) and Souri et al. (2016). Overall, the slope we report in our study (-0.68 ± 0.27 ppb y$^{-1}$) is larger but more certain compared to the slopes reported by Berlin et al. (2013), which were quantified regardless of the WD (-0.33 ± 0.39 ppb y$^{-1}$ and -0.21 ± 0.39 ppb y$^{-1}$). Compared to the value reported by Berlin et al. (2013), which represents the trend associated with SE winds only (-0.92 ± 0.74 ppb y$^{-1}$ or -0.75 ± 0.55 ppb y$^{-1}$), our slope derived from Approach B is smaller but twice as certain (-0.68 ± 0.27 ppb y$^{-1}$) and compares better with that reported by Souri et al. (2016) in terms of absolute error (-1.1 ± 0.55 ppb y$^{-1}$). Overall, the slopes from different approaches in this study and other studies are not significantly different (Fig. 8). The average background $O_3$ in this study is slightly larger (by 2-4 ppb) compared to that reported by Berlin et al. (2013), in any of the approaches except for the hourly median approach, which is smaller by up to 5 ppb. However, compared to Souri et al. (2016) the average estimates from our study and Berlin et al. (2013) are all much smaller, with differences ranging from 10 to 69 ppb (Table 5).

Both upper and lower background $NO_x$ also declined in all approaches, with significant slopes (see Table 5). No other long-term background $NO_x$ studies exist, making comparison impossible. Additionally, there is no long-term and season-scale evidence on the effect of $NO_x$ conversion to PAN and $HNO_3$ that could affect its temporal decline. Considering that the majority of the sites used to derive background $NO_x$ are urban sites or sites that are affected by fresh emissions, we could assume that conversion to PAN and $HNO_3$ might have had a minor effect on the temporal trends in background $NO_x$ and at the 6-months scale. However, we estimated a bias of ca. 30 % due to detection of PAN, $HNO_3$ and other nitrogen species as $NO_x$ (see p.15-16 in the SI). This, combined with the bias due to 8-h averaging of $NO_x$, has shifted the annual trends to lower ranges by 2 ppb. Regional background contributions to measured MDA8 $O_3$ are consistent with previously reported

contributions from Berlin et al. (2013), with the closest estimate of slope values spanning unity (from linear regression of measured MDA8 versus regional background) resulting from the approaches in which chemistry and meteorology were co-varied spatially and temporally; a higher estimate of slope value (by a factor of 5) resulted from the approach in which MDA8 $O_3$ was not constrained by $NO_x$ and meteorology.

## 4 Conclusions

The overall goals of this study were to estimate regional background $O_3$ and $NO_x$ in the HGB area and to quantify their temporal trends over the past decades. To design more efficient controls on local pollution, we need an improved understanding of regional contributions from a long-term perspective, and also better constraints on $O_3$ mixing ratio. We used up to seventeen years of hourly measurements of $O_3$ and $NO_x$ mixing ratios in different multivariate analysis approaches, including one that allowed covariance of $O_3$ with $NO_x$ and meteorology (T, WD and WS). Because we used ground-monitoring data, both background $O_3$ and $NO_x$ determined in this study represent the ground-level backgrounds, describing influences from regional chemistry and transport.

We found that the observed decline in regional background $O_3$ is real and quantifiable, regardless of the approach used to analyze the changes in regional background $O_3$ on the longest term possible. This is consistent with results from two previous studies (Berlin et al., 2013; Souri et al., 2016). Similarly, we detected and quantified a decline in the upper and lower bounds of background $NO_x$ in all approaches.

By accounting for the space-time covariance of $O_3$ with $NO_x$ and meteorology, we could better resolve the temporal trend of background $O_3$, with a more significant slope and improved coefficient of determination ($R^2$ of 0.62-0.63) on both time scales: 17 years and 13 years, respectively. Similarly, the temporal trend of background $NO_x$ resulted in a better performance of the linear model ($R^2 = 0.58$ compared to $R^2 = 0.53$) when the covariance of variables was used for the longest term, although the associated slope decreased slightly.

Our findings support the claim of Berlin et al. (2013) that changes in regional background $O_3$ also contributed to a local decline in MDA8 $O_3$. However, in our study, regional contributions to average MDA8 $O_3$ are underestimated when the space-time covariance of meteorology and chemistry is not considered (Fig. S16 vs. Fig. S18). When this covariance is accounted for in the analysis (our Approach B), the associated temporal trend in background $O_3$ (or $NO_x$) reflects both the effects of controlling precursor emissions and changes in meteorology. For instance, local chemistry was much more important in earlier years (prior to 2007) due to high emissions of $O_3$ precursors from petrochemical facilities, making it difficult to extract the regional background from surface data during those years. The trend became steadier after 2007 probably as an effect of emissions controls and a prevailing S-SE flow; this latter is consistent with the observed increased frequency of the southerly flow from the GOM (Liu et al., 2015). Based on a previous study (Wang et al., 2016), variations in the intensity and location of the Bermuda High could also explain some of the temporal behavior in summertime MDA8 $O_3$, causing a drop in mid-July, when southerly flow from the GOM is allowed to enter the region; this is marine background

$O_3$ and also contributes to the decline in regional background $O_3$ over time. We also observed this effect in regional background $O_3$ during July, particularly when using the hourly median approach.

Our estimates of 8-h based average background $O_3$ and $NO_x$ are both slightly overestimated compared to the hourly median approach, likely due to constraining the 8-h average $NO_x$ (and meteorology) by the MDA8 $O_3$. Future studies might consider refining these estimates by using a smaller time-averaging scale for $NO_x$, $O_3$ and meteorology. Although we estimated a bias of 18% due to 8-h averaging of $NO_x$, future refinements of background $NO_x$ would probably reduce this bias. In addition, corrections of $NO_x$ measurements that are representative for the region and the time periods analysed in this study are highly recommended to further improve the lower bound estimate of background $NO_x$; the average value of ca. 4 ppb still appears to be large compared to the short-term aircraft 'non-plume' $NO_x$ of 1-1.5 ppb observed in the region.

To test the linearity of the temporal trends in background $O_3$ and $NO_x$ and to continuously determine the effectiveness of control measures, and identify regulatory changes that need to be made, new studies should extend the trends in this study into future years. Additionally, wherever VOCs data are available, the extraction of background $O_3$ and $NO_x$ should be constrained over that period by VOCs as well and possibly by solar radiation. The related temporal trends should be compared over that period with those estimated from this study to highlight the effect of including VOCs and an additional meteorological variable in the multivariate analysis. Coincident solar radiation and $NO_x$ could also be used to test the conversion of $NO_x$ to oxidation products (PAN, $HNO_3$, etc.) and asses the magnitude of this effect on the declining background $NO_x$ in the HGB region.

**Author contribution**

L. G. Suciu (data collection and processing, data analysis and interpretation, manuscript writing); R. J. Griffin (guidance on data analysis and interpretation, critical revision of the manuscript); C. A. Masiello (critical revision of the manuscript).

**Data availability**

Time series of data analyzed in this study (validated raw data reports, JMP) are available at the Texas Air Monitoring and Information System (TAMIS) website owned by Texas Commission on Environmental Quality. The website can be accessed at: http://www17.tceq.texas.gov/tamis/index.cfm?fuseaction=home.welcome

**Competing interests**

The authors declare that they have no conflict of interest.

## Acknowledgments

This work was supported by the Texas Commission on Environmental Quality. Thanks to research scientist N. P. Sanchez (Department of Civil and Environmental Engineering, Rice University) for insight on PCA.

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

**Table 1: The $O_3$ and $NO_x$ sites and their loadings associated with each principal component using the hourly median approach**

| Site name | PC1 $O_3$ | PC1 $NO_x$ | PC2 $O_3$ | PC2 $NO_x$ | PC3 $O_3$ | PC3 $NO_x$ | PC4 $O_3$ | PC4 $NO_x$ | PC5 $O_3$ | PC5 $NO_x$ |
|---|---|---|---|---|---|---|---|---|---|---|
| Channelview | .714 | .161 | .501 | .905 | .233 | .075 | .367 | -.042 | N/A | .156 |
| Clinton | .830 | -.224 | .387 | .178 | .326 | .050 | .130 | .923 | N/A | -.005 |
| Conroe | -.084 | -.088 | .089 | .382 | .878 | .794 | -.188 | .005 | N/A | .235 |
| Conroe Relocated | .273 | .212 | -.183 | .233 | .900 | .700 | .076 | .530 | N/A | -.296 |
| Danciger | .969 | .841 | -.166 | .112 | .045 | .103 | .076 | -.425 | N/A | -.007 |
| Galveston 99 St. | .925 | .951 | -.279 | .020 | .057 | .190 | .044 | -.031 | N/A | -.062 |
| Galveston Airport | .960 | .974 | .100 | .043 | -.022 | .052 | -.133 | -.020 | N/A | -.013 |
| Houston Aldine | .373 | .413 | .549 | .788 | .712 | .368 | .193 | -.011 | N/A | .043 |
| Bayland Park | .856 | .837 | .272 | .387 | .390 | .260 | .046 | -.073 | N/A | .192 |
| Houston Crawford | -.055 | .835 | .906 | .441 | .223 | -.126 | -.063 | -.140 | N/A | -.003 |
| Deer Park | .881 | .871 | .369 | .402 | .274 | .181 | .067 | -.051 | N/A | .045 |
| Houston East | .460 | .918 | .577 | .341 | .552 | .064 | .290 | -.069 | N/A | .103 |
| Hayden Rd. (HRM3) | .765 | .324 | .481 | .780 | .334 | .477 | .205 | -.014 | N/A | -.013 |
| Sheldon Rd. (HRM4) | .044 | -.061 | .921 | .835 | .142 | .213 | .014 | .082 | N/A | -.154 |
| Baytown (HRM7) | .129 | -.451 | .952 | .135 | -.024 | -.187 | .008 | -.057 | N/A | .749 |
| La Porte (HRM8) | .405 | .444 | -.336 | .034 | -.131 | .159 | .641 | .009 | N/A | .782 |
| Mont Belvieu (HRM10) | -.141 | -.736 | .914 | .394 | .035 | .044 | -.237 | .311 | N/A | .257 |
| East Baytown (HRM11) | .035 | -.727 | .891 | .350 | -.174 | .124 | -.102 | -.094 | N/A | -.042 |
| Lynchburg Ferry | .827 | .382 | .410 | .773 | .156 | .090 | .194 | .097 | N/A | -.040 |
| Lake Jackson | .978 | .771 | -.157 | .207 | -.010 | .415 | .037 | -.340 | N/A | -.193 |
| Manvel Croix | .966 | .847 | -.021 | .336 | .223 | .346 | .087 | -.092 | N/A | -.127 |
| Mustang Bayou | .977 | .917 | -.162 | .209 | .065 | .149 | .011 | .152 | N/A | -.056 |
| NW Harris | .653 | .567 | .072 | .499 | .721 | .576 | -.056 | .121 | N/A | -.120 |
| Park Place | .901 | .829 | .228 | .484 | .311 | .223 | .148 | -.044 | N/A | .029 |
| San Jacinto Monument | .553 | .808 | .544 | .301 | -.029 | .004 | -.557 | -.120 | N/A | .160 |
| Seabrook Fr. Park | .971 | .931 | .085 | .236 | .176 | .164 | .064 | .103 | N/A | -.008 |
| Texas City 34 St. | .982 | .652 | -.104 | .223 | .049 | .632 | .017 | -.178 | N/A | -.108 |
| Wallsville Rd. | .849 | .171 | .406 | .829 | .123 | .077 | .142 | .263 | N/A | .313 |

**Table 2: The loadings or correlations of the components with variables at each site from Approach A**

| Site name | PC1 | | PC2 | | PC3 | | PC4 | | PC5 | |
|---|---|---|---|---|---|---|---|---|---|---|
| | $O_3$ | $NO_x$ | $O_3$ | $NO_x$ | $O_3$ | $NO_x$ | $O_3$ | $NO_x$ | $O_3$ | $NO_x$ |
| Houston Aldine | 0.609 | 0.172 | 0.516 | 0.209 | 0.333 | 0.940 | 0.259 | 0.127 | 0.430 | 0.163 |
| Bayland Park | 0.370 | 0.209 | 0.411 | 0.208 | 0.445 | 0.142 | 0.694 | 0.884 | 0.123 | 0.332 |
| Deer Park | 0.305 | 0.949 | 0.268 | 0.067 | 0.865 | 0.167 | 0.272 | 0.177 | 0.109 | 0.185 |
| Houston East | 0.775 | 0.227 | 0.380 | 0.173 | 0.382 | 0.194 | 0.320 | 0.347 | 0.079 | 0.872 |
| NW Harris | 0.371 | 0.067 | 0.814 | 0.950 | 0.301 | 0.203 | 0.310 | 0.175 | 0.114 | 0.144 |

**Table 3: The loadings or correlations of the components with variables at each site from Approach B**

| Site name | PC1 | | | | | PC2 | | | | |
|---|---|---|---|---|---|---|---|---|---|---|
| | $O_3$ | $NO_x$ | T | WD | WS | $O_3$ | $NO_x$ | T | WD | WS |
| Houston Aldine | 0.065 | -0.794 | 0.802 | 0.310 | 0.223 | 0.813 | 0.183 | 0.319 | -0.107 | -0.771 |
| Bayland Park | 0.805 | 0.463 | 0.267 | -0.160 | -0.787 | -0.075 | -0.698 | 0.810 | 0.541 | 0.057 |
| Deer Park | 0.820 | 0.648 | 0.123 | -0.159 | -0.779 | 0.053 | -0.549 | 0.929 | 0.330 | 0.167 |
| Houston East | 0.118 | -0.823 | 0.798 | 0.439 | 0.295 | 0.804 | 0.200 | 0.344 | -0.284 | -0.763 |
| NW Harris | 0.825 | 0.498 | 0.147 | -0.508 | -0.605 | 0.097 | -0.573 | 0.892 | 0.278 | -0.013 |

**Table 4: The loadings or correlations of the components with variables at each site from Approach C**

| Site name | PC1 | | | | | PC2 | | | | |
|---|---|---|---|---|---|---|---|---|---|---|
| | $O_3$ | $NO_x$ | T | WD | WS | $O_3$ | $NO_x$ | T | WD | WS |
| Houston Aldine | 0.780 | 0.319 | 0.145 | -0.243 | -0.804 | 0.236 | -0.773 | 0.835 | 0.086 | 0.127 |
| Bayland Park | 0.807 | 0.481 | 0.288 | -0.203 | -0.772 | -0.031 | -0.684 | 0.823 | 0.461 | 0.124 |
| Deer Park | 0.821 | 0.554 | 0.161 | -0.392 | -0.701 | -0.030 | -0.681 | 0.886 | -0.168 | 0.358 |
| Houston East | 0.272 | -0.794 | 0.859 | 0.223 | 0.155 | 0.736 | 0.344 | 0.149 | -0.399 | -0.814 |
| NW Harris | 0.784 | 0.451 | 0.130 | -0.535 | -0.668 | 0.082 | -0.697 | 0.900 | 0.159 | 0.060 |
| Channelview | 0.625 | 0.484 | 0.047 | 0.271 | -0.843 | 0.106 | -0.627 | 0.709 | 0.567 | -0.030 |
| Conroe Relocated | 0.741 | 0.560 | -0.007 | -0.015 | -0.844 | -0.207 | -0.664 | 0.723 | 0.666 | -0.139 |
| Manvel Croix | -0.825 | 0.625 | -0.042 | 0.627 | 0.717 | 0.103 | 0.065 | 0.941 | 0.510 | 0.074 |
| Clinton | -0.220 | 0.117 | 0.254 | 0.694 | 0.785 | 0.792 | 0.035 | 0.736 | 0.007 | -0.016 |
| Seabrook Fr. Park | 0.480 | 0.871 | -0.602 | 0.278 | -0.451 | -0.578 | -0.160 | -0.040 | 0.833 | 0.634 |

**Table 5: Comparison between all approaches in this study and literature**

| Method | Average regional background | | Temporal trends in regional background | | | |
|---|---|---|---|---|---|---|
| | $O_3$ | $NO_x$ (or $NO_2$) | $O_3$ | | $NO_x$ | |
| | ppb | ppb | Slope (ppb y$^{-1}$) | $R^2$ | Slope (ppb y$^{-1}$) | $R^2$ |
| Approach A (17 years) | $46.74 \pm 0.58$[†] | $6.86 \pm 0.19$[†] | $-0.13 \pm 0.10$ | 0.27 | $-0.06 \pm 0.03$ | 0.53 |
| | | $4.49 \pm 0.12$[ℓ] | | | $-0.04 \pm 0.02$[ℓ] | 0.53[ℓ] |
| Approach B (17 years) | $46.72 \pm 2.08$[†] | $6.80 \pm 0.13$[†] | $-0.68 \pm 0.27$ | 0.63 | $-0.04 \pm 0.02$ | 0.58 |
| | | $4.45 \pm 0.08$[ℓ] | | | $-0.03 \pm 0.01$[ℓ] | 0.58[ℓ] |
| Approach C (13 years) | $44.71 \pm 1.28$[†] | $6.03 \pm 0.05$[†] | $-0.49 \pm 0.24$ | 0.62 | $-0.013 \pm 0.012$ | 0.30 |
| | | $3.95 \pm 0.03$[ℓ] | | | $-0.009 \pm 0.008$[ℓ] | 0.30[ℓ] |
| Hourly median (up to 17 years) | $37.60 \pm 1.55$[*] | $5.75 \pm 0.62$[*] | | | | |
| | | $4.05 \pm 0.44$[ℓ] | | | | |
| Adjusted hourly median (up to 17 years) | $37.67 \pm 0.80$[§] | $5.74 \pm 0.32$[§] | | | | |
| | | $4.03 \pm 0.09$[ℓ] | | | | |
| Berlin et al. (2013) (14 years) | $42.5 \pm 6.3$[Ł] | | $-0.33 \pm 0.39$ | 0.23 | | |
| | | | $-0.21 \pm 0.39$ | 0.12 | | |
| | | | $-0.92 \pm 0.74$[Ł] | | | |
| | | | $-0.79 \pm 0.65$[Ł] | | | |
| Souri et al. (2016) (15 years) | $107 \pm 27$[ς] | $(10 \pm 3)$[ς] | $-1.0 \pm 0.5$[ς] | | | |
| | $77 \pm 27$[þ] | $(8 \pm 3)$[þ] | $-0.9 \pm 0.86$[þ] | | | |
| | $57 \pm 19$[ρ] | $(6 \pm 3)$[ρ] | $0.09 \pm 0.40$[ρ] | | | |

[†]The average values were obtained by averaging the yearly values over the respective study period; the yearly values represent the season means (May-Oct) and account for daytime hours only.

[*]The hourly background values (daytime hours during May-Oct) were averaged over 8 hours for each month to get the season mean that is comparable with the other approaches. This background is based on a single component (PC1).

[§]The hourly background was adjusted to include average regional contributions from two components (PC1 and PC3).

[Ł] Constrained by wind direction from southeast

[ς] Constrained by wind direction from east-northeast

[þ] Constrained by wind direction from east-southeast

[ρ] Constrained by wind direction from south-southeast

() Regional background $NO_2$ (average of both daytime and nighttime)

[ℓ] Lower bound of background $NO_x$ (corrected for time-averaging and/or measurement bias, see p.15-16 in the SI).

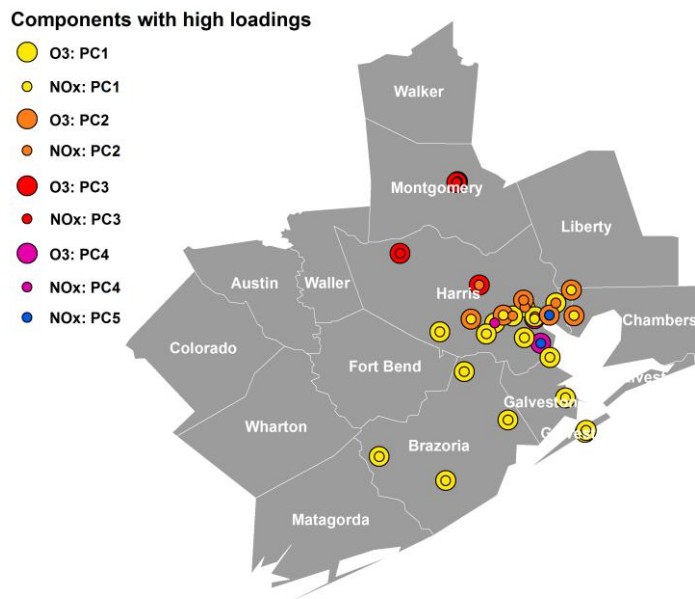

**Figure 1: Distinct clustering of principal components. The cluster in yellow is PC1-O$_3$ and PC1-NO$_x$. The cluster in orange is PC2-O$_3$ and PC2-NO$_x$, and so on. Smaller circles represent NO$_x$ clusters.**

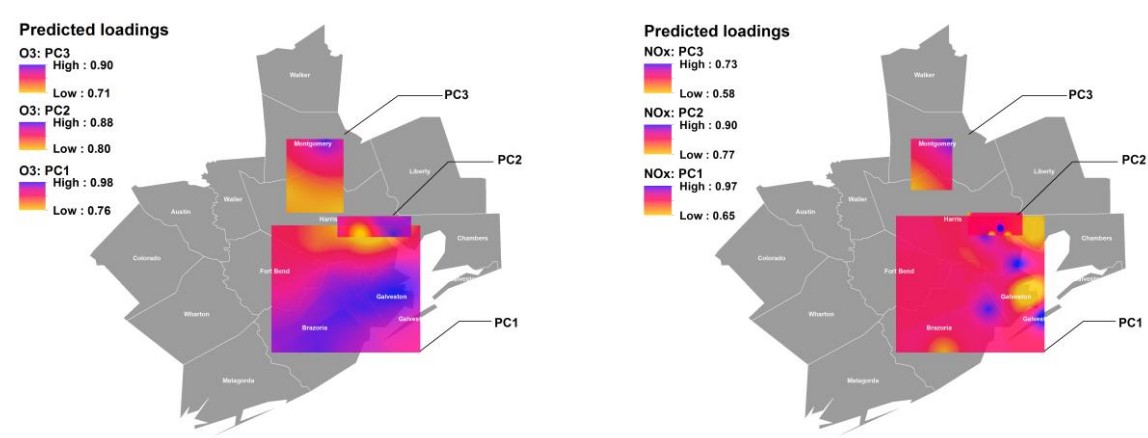

**Figure 2: Spatial interpolation of normalized squared loadings from the highly correlated sites with the first three components in terms of O$_3$ (left) and NO$_x$ (right). Range is from 0 to 1.**

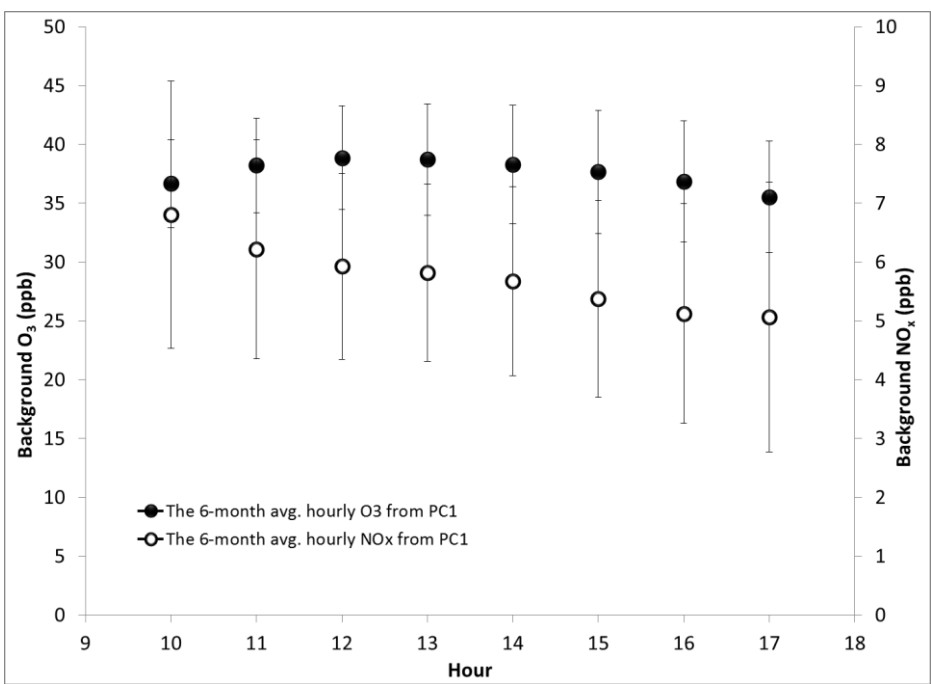

**Figure 3: The season averaged hourly background O₃ and hourly background NOₓ. Error bars represent the 95% confidence interval for the mean.**

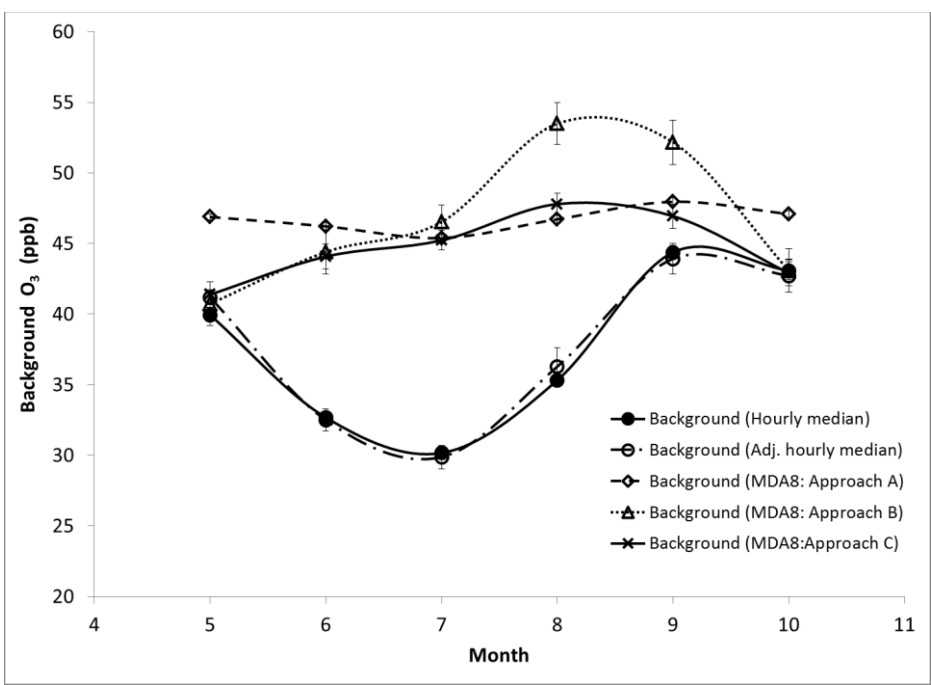

**Figure 4: The 6-month trends in background O₃ from different approaches. Points represent the monthly average background values derived from the hourly median O₃ and MDA8 O₃. Error bars represent the 95% confidence interval for the mean.**

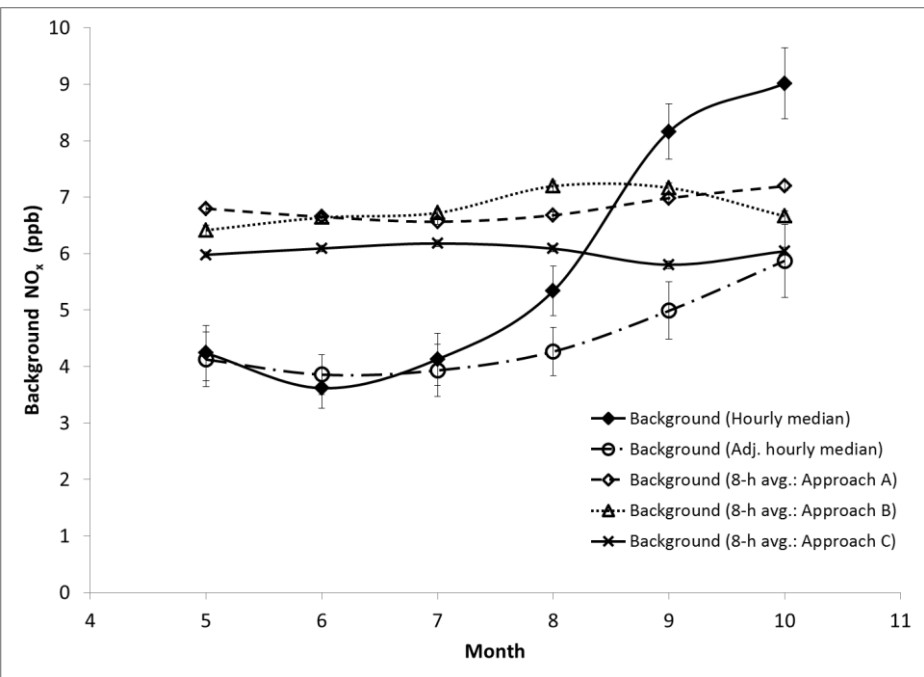

**Figure 5: The 6-months trends in background NO$_x$ from different approaches. Points represent the monthly average background values derived from the hourly median NO$_x$ and the 8-h average NO$_x$. Error bars represent the 95% confidence interval for the mean.**

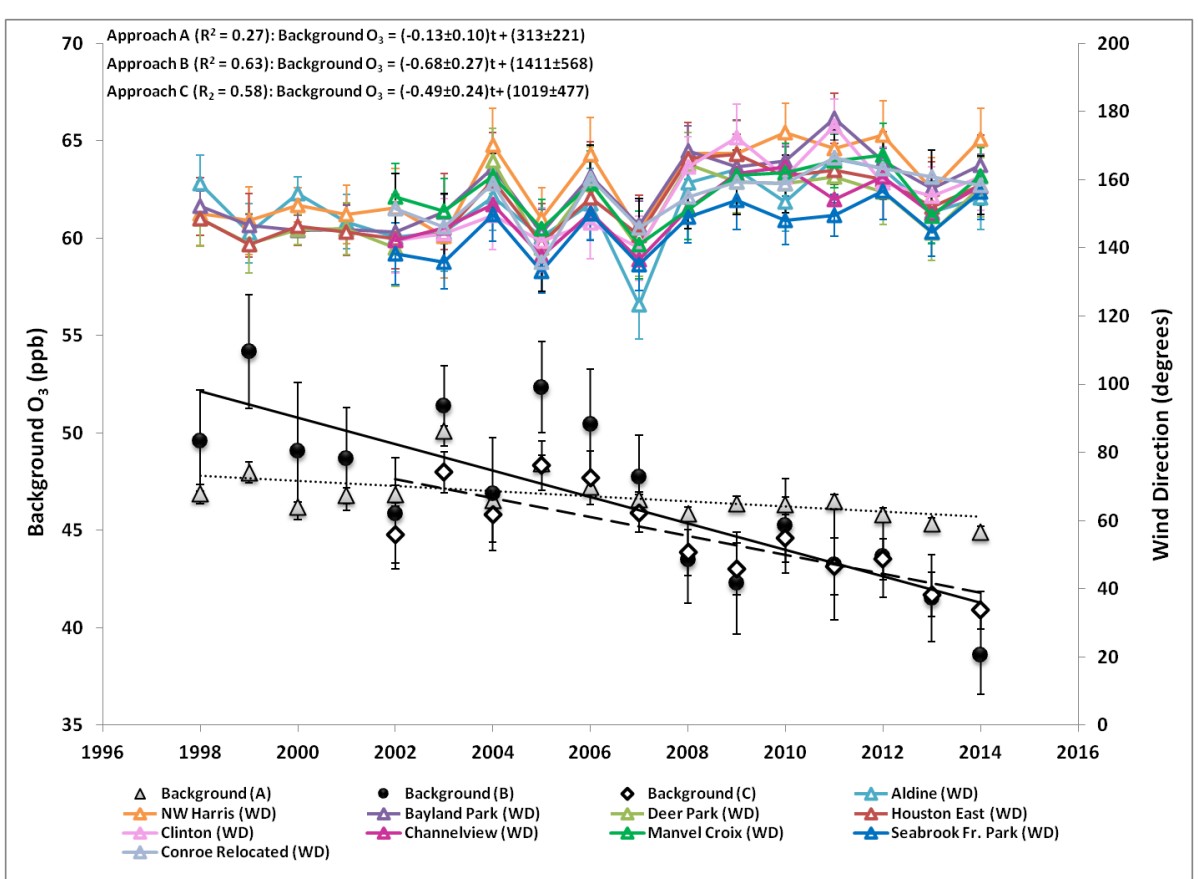

**Figure 6: Temporal trends in background O₃ (Approaches A-C) and average wind direction. Error bars represent the 95% confidence interval for the mean.**

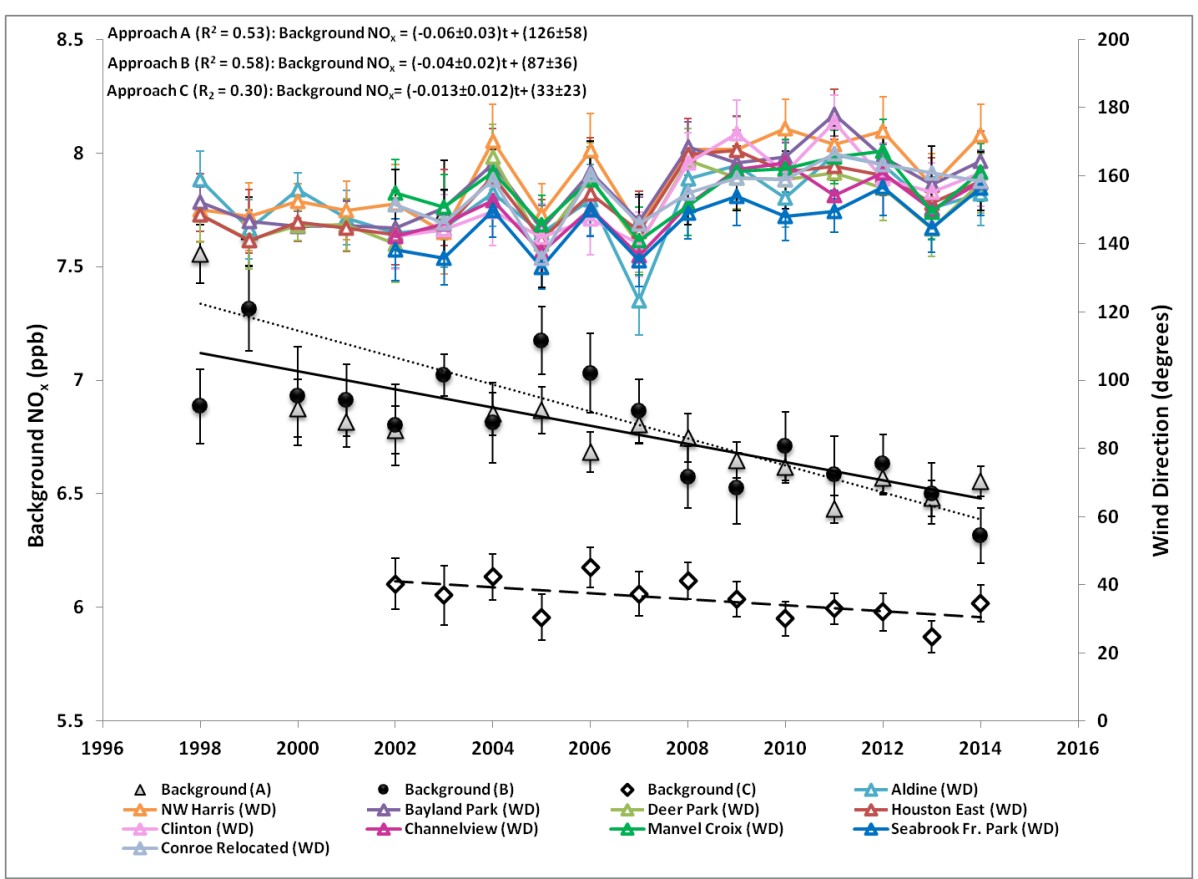

**Figure 7: Temporal trends in upper bound background NO$_x$ (Approaches A-C) and average wind direction at various sites. Error bars represent the 95% confidence interval for the mean.**

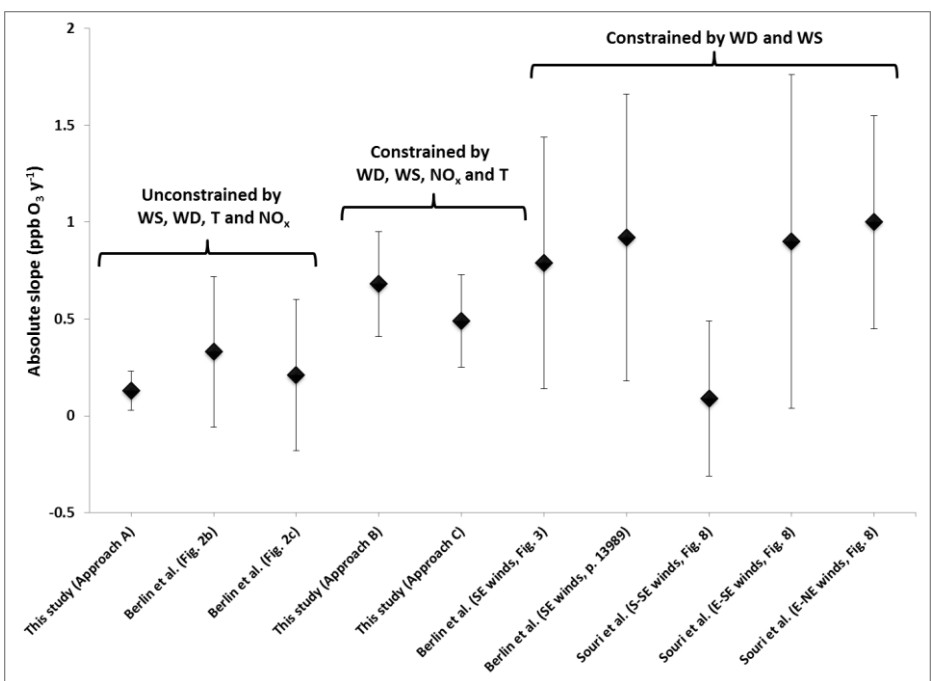

**Figure 8: Comparison between the slopes of temporal trends in regional background O$_3$ in the HGB region.**