# Peer review of "Regional background O\_3 and NO\_x in the Houston-Galveston-Brazoria (TX) region: A decadal-scale perspective"

_Atmospheric Chemistry and Physics, 2016_

## Referee Comment (RC1) · Anonymous Referee #1 · 28 Nov 2016

Review of ACP-2016-893

General Comments

This paper describes the use of principal component analysis (PCA) to examine trends in the background concentrations of O3 and NOx in the greater Houston, TX area over a 17 year period (for ozone). Understanding the significant decreases that have occurred in this former extreme non-attainment area since 2000 can help us better understand the effectiveness of ozone control strategies in general. The authors have expanded on the work of previous researchers that applied this technique to maximum daily 8-h average (MDA8) ozone concentrations, by extending the PCA analysis to include NOx and adding 1-h median values of both O3 and NOx to their analysis. The

paper is well written and easy to read. The results, which are consistent with the earlier findings, are both useful and important and should be published with minor revisions to address the points raised below.

Specific Comments.

P3, L16. The authors should point out that one of the key findings of the first Tex-AQs study was the disproportionate role of highly reactive VOCs (HRVOCs), primarily alkenes, released from petroleum refineries in the rapid production of ozone in the Houston area. These "upset" emissions were greatly reduced before the second study took place, greatly reducing the local ozone contributions.

P3, L25-29. The authors should consider including the study of Darby et al. in their introduction.

Darby, L. S. (2005), Cluster analysis of surface winds in Houston, Texas, and the impact of wind patterns on ozone, Journal of Applied Meteorology, 44(12), 1788-1806, doi:1710.1175/JAM2320.1781.

P6, L8. Did the Varimax rotation make any difference in the interpretation compared to the unrotated PCs?

P6, L18. A logical extension of this work would be to apply the PCA techniques to the diurnal 1-h median values of Ox (=O3+NOx), which is more conservative. Indeed, an analysis of the nighttime Ox concentrations when there is no photochemical activity might provide the best estimate of background ozone.

P7, L26. As noted later in the paper, PC1 equally represents the marine and continental backgrounds depending on the sign.

P8, L6. Figure 2 suggests that the primary NOx PC loadings are associated with the W.A. Parish and other power plants; is this the case?

P8, L29. Another explanation for the difference is that much of the NOx responsible for

the background O3 production has been converted to NOy (e.g. HNO3 and PAN). This would include most lightning generated NOx. Also, depending on the season, a significant amount of the background ozone may also have originated from the stratosphere.

P9, L30. See comment above about VOCs and Daum et al. (for example)

Daum, P. H., L. I. Kleinman, S. R. Springston, L. J. Nunnermacker, Y.-N. Lee, J. Weinstein-Lloyd, J. Zheng, and C. M. Berkowitz (2004), Origin and properties of plumes of high ozone observed during the Texas 2000 Air Quality Study (TexAQS 2000), J. Geophys. Res, 109, doi:10.1029/2003JD004311.

P11, L12. PC5 is not significant for O3.

P13, L25+. The variations in Fig. 4 suggest that the 1-h median approach is (not surprisingly) more strongly influenced by the persistent onshore flow during July than the 8-h MDA8 approach.

P16, L10. The slope is listed as -0.68±0.27 in Table 7.

P16, L13. The slopes all agree within the margins of error and are not significantly different.

P16, L15. The background ozone trend estimates derived from the current analysis may be twice as precise as those in Berlin et al. (2013), but they are not necessarily more accurate. Indeed, the large interannual variations in the method B data plotted in Figures 6 and 7 suggest that a linear model is not really appropriate. Some discussion of this is in order.

Technical Corrections

P1, L8 (Abstract). Suggest omitting the "the" to give: "...photochemistry is most active..."

P1, L24 (Abstract). Suggest replacing the "the" to give: "...since 2007 and an increase in..."

P3, L16. "Parrish" is misspelled in the reference.

P9, L24. What does VOCs mean? Is this a typo?

---

## Referee Comment (RC2) · Anonymous Referee #2 · 3 Dec 2016

The manuscript "Regional background O3 and NOx in the Houston-Galveston-Brazoria (TX) region: A decadal-scale perspective" by Suciu et al. determines the composition of air coming in to the region using a principal component analysis of O3 and NOx monitoring data. The results for background O3 presented here are consistent with previous studies that used similar data and analysis. New contributions include consideration of long-term changes in meteorology and in background NOx. The manuscript is well-organized has been edited carefully.

I am uncertain about the meaning of much of the analyses. One primary concern is that the analysis of background NOx is incomplete or possibly in error. In this manuscript, NOx is averaged over 8 h periods corresponding to the maximum daily average 8 hr

ozone. But NOx and ozone do not have the same temporal behavior, so I don't think this average can be used to determine background NOx. NOx is usually greatest at very different times than ozone. Although NOx is important to O3, the two often anti-correlate. So this analysis could miss large NOx values that occur earlier in the day.

NOx is defined in the introduction to be the sum of NO and NO2. But the monitoring NOx reported here is from chemiluminescence detectors with a molybdenum converter that also detects PAN and some HNO3. This limitation isn't critical for measurements in urban regions dominated by fresh emissions, but the contributions from PAN, HNO3, and other oxidized reactive nitrogen compounds is likely substantial if background locations and times are considered. The meaning of monitoring NOx has been discussed in many papers (e.g. , Winer et al., Response of Commercial Chemiluminescent NO-NO, Analyzers to Other Nitrogen-Containing Compounds, ES&T, 1974), and it should be considered here. If monitoring NOx is used to examine background levels, there needs to be considerably more examination of the data, and it may be impossible to use NOx for this sort of analysis. For example, all of the trends in NOx could be dominated by changes in partitioning between the NOx oxidation products (PAN, organic nitrates, and HNO3), rather than a reduction in NOx. If the ratio of organic nitrogen to HNO3 has changed in the background air (which is likely), then the monitoring NOx instruments would likely respond in a way that would alter trends in NOx.

The background NOx value of 6.8 ppbv is surprisingly large, and it is inconsistent with the 2000 and 2006 intensive field studies that showed NOx upwind of HGB was often <1 ppbv, and NOy was a 1-4 ppbv (see for example the upwind or non-plume measurements shown in Daum, P. H., et al., A comparative study of O3 formation in the Houston urban and industrial plumes during the 2000 Texas Air Quality Study, J. Geophys. Res., 108(D23), 4715, doi:10.1029/2003JD003552, 2003; Ryerson, T. B., et al. (2003), Effect of petrochemical industrial emissions of reactive alkenes and NOx on tropospheric ozone formation in Houston, Texas, J. Geophys. Res., 108(D8), 4249, doi:10.1029/2002JD003070; Neuman, J. A., et al., Relationship between photochemical ozone production and NOx oxidation in Houston, Texas, J. Geophys. Res., 114, D00F08, doi:10.1029/2008JD011688). I don't know whether the discrepancy is an artifact of the data or the analysis, or both. But if background NOx were truly 6.8 ppbv, then NOx emission controls in HGB would need to be reconsidered. NOx is short-lived, and it is possible that the NOx measured at these monitoring stations is strongly influenced by local emissions. I recommend removing the analysis of background NOx, or adding substantial discussion and examination of the NOx data.

The use of MDA8 needs to be put into context, and the importance of MDA8 should be discussed. MDA8 is a regulatory construct. Is HGB in exceedance of the O3 standard? What is the current O3 standard (only the old standard is mentioned)? It would be helpful to indicate the NAAQS on the figures. The background fraction of total ozone discussed in section 3.7 also has me confused, and I think it misses the point of MDA8. The background MDA8 is important insofar as it contributes to the design value for the entire air basin. So background MDA8 should be compared with the largest MDA8 in the region to understand the effect of the background on compliance with O3 regulation. I don't see the point of comparing background MDA8 to an average of MDA8 from the same locations, as shown in figures S18-25. If the analysis finds sites and conditions that faithfully represent the background, then shouldn't the background MDA8 always equal the measured MDA8? Why are there so many points in the supplementary figures with the PCA-derived background O3 greater than the measured O3?

Some of the language is imprecise, and I had to read the sentences many times to distinguish the literal meaning from the authors' likely intent. For example, pg 2 line 14 states that "no study has yet to quantify the regional contributions to direct O3 precursors themselves...". I'm not sure what this means. Zhang et al, and many other papers, examines background O3 precursors, and is already referenced. The second sentence of the abstract states that ozone dependence on VOC:NOx ratio makes ozone difficult to control locally. I think the whole point of this paper is that large background contributions, rather than the VOC:NOx, may make local ozone control challenging. I don't

understand page 3, line 1 that says "Meteorological controls . . .are reflected by a more significant decline .. in the east than in the west". Cooper et al explain this difference by changes in Asian emissions and biomass burning, not changes in meteorology. I don't understand page9, line 19: why does NOx increase with windspeed? The last paragraph of the conclusion is not supported by the manuscript. Rather than emphasize work that needs to be performed, the authors should focus on their most important findings.

The results reported here can be made more valuable by further synthesizing the findings. There are 25 figures in the Supplementary section, and it is hard to distinguish one from the next. The first 6 tables are very dense, showing many PCA loadings for many different sites. I don't think many readers will be able to use all these tables of numbers and all the figures in the Supplementary. This paper examines many topics, and most points are supported by a scatter plot and the associated statistics obtained from a linear least squares fit. It is challenging to appreciate the important findings, as they are obscured by an abundance of data and statistics. An in-depth consideration of a single topic, such as the decadal change in wind direction and its effect on background ozone, would be a more powerful contribution to the literature.

---

## Author Response (AR1)

**Final author response and revisions of the manuscript: "Regional background O$_3$ and NO$_x$ in the Houston-Galveston-Brazoria (TX) region: A decadal-scale perspective" by Suciu et al.**

**Author response to Referee #1**

We thank Referee # 1 for the positive remarks about the significance of our study and easiness of reading, and for the helpful suggestions to improve our manuscript. Our answers to the referee's comments are listed below.

Specific comments

*Referee comment:*

P3, L16. The authors should point out that one of the key findings of the first Tex-AQs study was the disproportionate role of highly reactive VOCs (HRVOCs), primarily alkenes, released from petroleum refineries in the rapid production of ozone in the Houston area. These "upset" emissions were greatly reduced before the second study took place, greatly reducing the local ozone contributions.

*Author response:*

We agree with the statement made by the referee, which is consistent with previous studies, which were not included in our paper (Ryerson et al., 2003; Daum et al., 2004). We added the following on P3, L16-18:

> "The O$_3$ pollution in this region was likely a result of abundant precursors emitted locally from urban and industrial sources (particularly, the highly reactive VOCs (HRVOCs) from the petroleum refineries) and the local chemistry sustained by the high summer temperature and land-sea breeze effects. However, the emissions of HRVOCs have been considerably reduced after the first campaign, resulting in lower local contributions to O$_3$."

We also included two more studies in the cited references on P3, L16-17:

> "Two intensive air quality campaigns investigated peak O$_3$ in the HGB region during 2000 and 2006, respectively (Ryerson et al., 2003; Daum et al., 2004; Banta et al., 2005; Rappenglück et al., 2008; Parish et al., 2009; Pierce et al., 2009; Langford et al., 2010)."

*Referee comment:*

P3, L25-29. The authors should consider including the study of Darby et al. in their Introduction

*Author response:*

The short term cluster analysis on hourly wind and ozone maxima in the Houston area (Darby et al., 2005) is a good suggestion, not only for this line. It could also be added on P3, L14 as it points out that the transition from offshore to

onshore flow causes high $O_3$ concentrations (>140 ppb) on a 1-h basis, which is in the line with the land-sea breeze effects described in that paragraph. Therefore we included it among other studies on P3, L14:

> "The land-sea breeze effect complicates this picture through recirculation of local pollution and formation above the coast of the Gulf of Mexico (GOM) of stagnant air masses that entrain local precursors and favor local chemistry and formation of $O_3$ (Banta et al., 2005; Darby et al, 2005; Nielsen-Gammon et al., 2005; Rappenglück et al., 2008; Langford et al., 2009)."

Recently, we also found that Souri et al. (2016) did cluster analysis on 900 hPa winds and surface $O_3$ and reported long-term temporal trends in MDA8 $O_3$ by wind cluster. Darby et al. (2005) did something similar in terms of describing 1-h $O_3$ maxima by wind patterns but on a much shorter term. Therefore we made the following revision on P3, L22-28:

> "Regional background $O_3$ in the HGB region has been quantified by many studies but results vary, depending on the temporal scale, spatial scale and the altitude of observations used in data analysis (Banta et al., 2005; Darby et al., 2005; Nielsen-Gammon 2005; Rappenglück et al., 2008; Kemball-Cook et al., 2009; Langford et al., 2009; Zhang et al., 2011; Banta et al., 2011; Berlin et al., 2013; Liu et al, 2015; Souri et al., 2016). Most of the above studies used the MDA8 $O_3$ to quantify background $O_3$. Overall, regional (continental) background $O_3$ ranges from 16 to 107 ppb, while marine background has values between 18 and 40 ppb. Local $O_3$ contributions are between 25 and 80 ppb. Observations from 1-h average $O_3$ data and using wind patterns resulted in higher $O_3$ mixing ratios, particularly during stagnation in the afternoon (>140 ppb) (Darby et al., 2005). Meteorological variables, such as wind patterns, were used separately to characterize the transport regime and its diurnal transition in the HGB region and interpret their findings from data analysis; their covariance with $O_3$ and $NO_x$ was not considered."

*Referee comment:*

P6, L8. Did the Varimax rotation make any difference in the interpretation compared to the unrotated PCs?

*Author response:*

The rotation gave different loadings for each PC. Because the primary interpretation of the PCs was based on the loading values, yes it made a difference. For instance, in approach A (independent PCA on MDA8 $O_3$ and 8-h average $NO_x$ at 5 sites), the absolute values of the loadings for PC1 were all greater than 0.9, without rotation. However, when using the Varimax rotation, only 2 out of 5 sites had significant loadings (absolute values nearly or greater than 5). The situation is similar for $NO_x$ except that the loadings in PC1 were all greater than 0.6 (no rotation) and only 1 out of 5 sites had a significant loading value after rotation (0.95).

*Referee comment:*

P6, L18. A logical extension of this work would be to apply the PCA techniques to the diurnal 1-h median values of Ox (=O3+NOx), which is more conservative. Indeed, an analysis of the nighttime Ox concentrations when there is no photochemical activity might provide the best estimate of background ozone.

*Author response:*

Yes, it would be interesting to apply the PCA method to 1-h median $O_x$, which is defined by Daum et al. (2004) as being the sum of $O_3$ and $NO_2$, to estimate regional background $O_3$. We did not use it because we wanted to assess the relationship between regional $NO_x$ and regional $O_3$; this required independent analyses of 1-h median $O_3$ and $NO_x$. Moreover, due to the limitation of the measurement method, $NO_2$ might include other oxidation products (PAN, $HNO_3$, etc.). The nighttime background could also be the recirculated local pollution from the previous day and might be different than the "regional" background in the following day. Our focus was on daytime regional background because of its important contribution to peak $O_3$.

*Referee comment:*

P7, L26. As noted later in the paper, PC1 equally represents the marine and continental backgrounds depending on the sign.

*Author response:*

We agree that on the scale of the high $O_3$ season (May-Oct), both marine and continental influences are described by PC1 based on the sign of its loadings. However, the statement in this line is related to the proximity to the GOM of the high PC1 loadings. We rephrased the text on P7, L26 to read: "The proximity to the GOM emphasizes that PC1 is largely influenced by marine background during summer."

*Referee comment:*

P8, L6. Figure 2 suggests that the primary NOx PC loadings are associated with the W.A. Parish and other power plants; is this the case?

*Author response:*

The primary $NO_x$ PC loadings are the results of interpolating between the monitoring sites. Some monitoring sites are in the proximity of power plants. The W.A. Parish power plant is in the northwest of the PC1 pattern.

*Referee comment:*

P8, L29. Another explanation for the difference is that much of the NOx responsible for the background O3 production has been converted to NOy (e.g. HNO3 and PAN). This would include most lightning generated NOx. Also, depending on the season, a significant amount of the background ozone may also have originated from the stratosphere.

*Author response:*

We agree with the referee regarding $NO_x$ conversion to $NO_y$ but we don't think that lightning $NO_x$ and stratospheric O3 are important contributions in the HGB region based on available studies. Therefore, we added the following on P8, L31:

"It is also possible that a fraction of background $NO_x$ (including lightning $NO_x$) was converted to PAN and $HNO_3$, which was accounted for in the total $NO_x$ by the measurement method, reducing the potential of background $NO_x$ to

explain background $O_3$. Stratospheric $O_3$ also may explain some of the background $O_3$ in the HGB. However, stratospheric $O_3$ contributions are either overestimated at mid-latitudes by the global cross-tropopause transport models (Liu et al., 2016) or the relationship between the cosmogenic beryllium-7 associated with particulate matter and surface $O_3$ observed in the HGB region is not conclusive enough (Gaffney et al., 2005). Modelling based estimates of lightning $NO_x$ in the Gulf of Mexico suggest that this source is negligible near the surface, ranging from near zero to 50 ppt during two summer months (Pickering et al. 2016)."

*Referee comment:*

P9, L30. See comment above about VOCs and Daum et al. (for example)

Daum, P. H., L. I. Kleinman, S. R. Springston, L. J. Nunnermacker, Y.-N. Lee, J. Weinstein-Lloyd, J. Zheng, and C. M. Berkowitz (2004), Origin and properties of plumes of high ozone observed during the Texas 2000 Air Quality Study (TexAQS 2000), J. Geophys. Res, 109, doi:10.1029/2003JD004311.

*Author response:*

The referee points out the work of Daum et al. (2004), which provides support to our statement "The unexplained portion for the 1-h level (70%) is quite significant. We believe it is related to rapid VOC chemistry in this area of the HGB region."

We thank the referee for this suggestion. Therefore, we added the following on P9, L31:

"Daum et al. (2004) measured various plumes for almost two weeks in late summer of 2000 and showed that six of them were different from typical urban plumes: they were rich in formaldehyde and peroxides, attributable to hydrocarbon oxidation and photochemistry, respectively. They also found that $O_3$ formation in these plumes was very efficient (6.4-11 ppbv $O_3$/ppbv of $NO_x$). These plumes were tracked back to sources of $NO_x$ and hydrocarbons in the proximity of the Houston Ship Channel. Using zero-dimensional model predictions, they found that $O_3$ formed very fast (140 ppbv/h). Compared to urban plumes, the authors found that the formation of $O_3$ in plumes from the Ship Cannel was more $NO_x$-limited, but uncertainties remain whether the production of $O_3$ in this area is $NO_x$- or VOCs-limited."

*Referee comment:*

P11, L12. PC5 is not significant for O3.

*Author response:*

Indeed, PC5 is not significant for $O_3$ (eigenvalue less than 1).

We made the following change on P11, L12:

"However, we retained all five components because they were not significantly different in explaining the variance in the original variables, particularly for $NO_x$ (Table 3). PC5 was not significant for $O_3$."

*Referee comment:*

P13, L25+. The variations in Fig. 4 suggest that the 1-h median approach is (not surprisingly) more strongly influenced by the persistent onshore flow during July than the 8-h MDA8 approach.

*Author response:*

We added the following on P13, L25-28:

"In Fig. 4, the hourly median approach also reveals a stronger onshore effect than the MDA8 $O_3$ approach. This could be because of the smaller time scale of observations, which allows the median to capture better the influence of the onshore flow in terms of $O_3$."

We also added the following on P13, L30:

"Regardless of the approach, background $O_3$ drops in July, which is consistent with the bimodal variation of the annual 8-h average background $O_3$ (Nielsen-Gammon et al., 2005) and with the less intense and a more easterly Bermuda High during July (Wang et al., 2016)."

*Referee comment:*

P16, L13. The slopes all agree within the margins of error and are not significantly different.

*Author response:*

The slopes from different approaches in this study and other studies are not different given the error bars (Fig.1, below), but those from this study appear to have lower uncertainties, regardless of the approach.

[Figure]

**Figure 1: Comparison between the slopes of temporal trends in regional background O₃ in the HGB region.**

*Referee comment:*

P16, L15. The background ozone trend estimates derived from the current analysis may be twice as precise as those in Berlin et al. (2013), but they are not necessarily more accurate. Indeed, the large interannual variations in the method B data plotted in Figures 6 and 7 suggest that a linear model is not really appropriate. Some discussion of this is in order.

*Author response:*

We propose the following change on P16, L11-15:

"Overall, the slope we report in our study (-0.68 ± 0.27 ppb y⁻¹) is larger but more certain compared to the slopes reported by Berlin et al. (2013), which were quantified regardless of the WD (-0.33 ± 0.39 ppb y⁻¹ and -0.21 ± 0.39 ppb y⁻¹). Compared to the values reported by Berlin et al. (2013), which represent the trend associated with SE winds only (-0.92 ± 0.74 ppb y⁻¹ or -0.79 ± 0.65 ppb y⁻¹), our slope derived from Approach B is smaller but twice as certain (-0.68 ± 0.27 ppb y⁻¹) and compares better with that reported by Souri et al. (2016) in terms of absolute error (-1.0 ± 0.55 ppb y⁻¹)."

The linear model is appropriate despite the larger interrannual variation in the early years, particularly for approach B. These variations are probably due to the fact that we only used five sites or to the fact that local chemistry was much more important in earlier years due to high emissions of O₃ precursors from petrochemical facilities, making it difficult to extract the regional background from surface data during those years. We think that the spatial scale also has an effect on the

interannual variability and we see it in the slightly smaller error bars in Approach C, when meteorology and chemistry are covaried between twice as many sites as used in Approach B. Statistically, we cannot reject the linear model to quantify the temporal trends in background $O_3$ and $NO_x$ because the model parameters are significant. We also used the linear model because previous studies used it, and we wanted to be able to compare our trends. Physically, we agree that a linear model might not be appropriate because there are many confounding factors that influence background $O_3$ and $NO_x$ on the long term. However, we could not account for all these factors in our study to test for non-linearity.

Technical corrections

*Referee comment:*
P1, L8 (Abstract). Suggest omitting the "the" to give: "…photochemistry is most active…"

*Author response:*
We made the correction on P1, L8:

"Ozone ($O_3$) in the lower troposphere is harmful to people and plants, particularly during summer, when photochemistry is most active and higher temperatures favor local chemistry."

*Referee comment:*
P1, L24 (Abstract). Suggest replacing the "the" to give: "…since 2007 and an increase in…"

*Author response:*
We changed the following on P1, L23-24:

"This decline is likely caused by a combination of state of Texas controls on precursor emissions since 2007 and an increase in frequency of flow from the Gulf of Mexico over the same time period.

*Referee comment:*
P3, L16. "Parrish" is misspelled in the reference.

*Author response:*
We corrected it on P3, L16:

"Two intensive air quality campaigns investigated peak $O_3$ in the HGB region during 2000 and 2006, respectively (Banta et al., 2005; Rappenglück et al., 2008; Parrish et al., 2009; Pierce et al., 2009; Langford et al., 2010)."

*Referee comment:*
P9, L24. What does VOCs mean? Is this a typo?

*Author response:*

Yes, it is a typo. We made the correction on this line:

"However, the high scores in July and August might be related to $NO_x$ and VOCs chemistry, rather than vertical mixing due to a higher boundary layer."

**Author response to Referee #2**

We thank Referee #2 for reviewing our manuscript. Our answers to the referee's comments are given below.

*Referee comment*

I am uncertain about the meaning of much of the analyses. One primary concern is that the analysis of background NOx is incomplete or possibly in error. In this manuscript, NOx is averaged over 8 h periods corresponding to the maximum daily average 8 hr. But NOx and ozone do not have the same temporal behavior, so I don't think this average can be used to determine background NOx. NOx is usually greatest at very different times than ozone. Although NOx is important to O3, the two often anticorrelate. So this analysis could miss large NOx values that occur earlier in the day.

*Author response*

This is the first attempt to resolve background $NO_x$ in the HGB on the long term using surface data. Whether our background $NO_x$ analyses are incomplete or possibly in error can be answered by comparing our background $NO_x$ estimates to those from other studies. Unfortunately, there are no long-term studies on background $NO_x$. A two-week study (cited in our manuscript, Zhang et al., 2011), used modeling and surface observations to determine regional and local $NO_x$ source contributions to $O_3$ in the HGB. The regional "upwind" $NO_x$ contribution to daily average $O_3$ were estimated to 20-60 ppb, while those from neighbor states to 20-25ppb. However, these large estimates are not representative to the season scale and the time period we used in our analysis. A recent long-term study (Souri et al., 2016) report 1-h average daytime and nighttime $NO_2$ in the HGB in a range of 6-10 ppb, which is not directly comparable to our 1-h daytime background $NO_x$.

We agree that $O_3$ and $NO_x$ do not have the same temporal behavior (see P5, L10-12 and P15, L29-31).

Regarding the use of the 8-h average $NO_x$ corresponding to the MDA8 $O_3$ to estimate background $NO_x$, please see our statement on P15, L31. We could also look at Fig. 5 in the manuscript and compare the 8-h average background $NO_x$ corresponding to MDA8 $O_3$ (Approaches A-C) with the adjusted hourly background $NO_x$ unconstrained by 1-h peak $O_3$ (the hourly median approach). The background $NO_x$ from the hourly median approach, averaged over 8 daytime hours for comparison, is lower by 1-2 ppb than the 8-h average background $NO_x$ from mid-July to early September, when important local chemistry is expected. This observation supports our statement on P15, L31 and suggests that we likely overestimate background $NO_x$ when the 8-h average $NO_x$ corresponding to MDA8 $O_3$ is used in the analysis.

We also agree that $O_3$ and $NO_x$ are anticorrelated if there is significant chemistry between them. However, our primary focus here was to find the opposite behavior in the extracted regional background $O_3$ and $NO_x$, which we see in Fig. 3 of the

manuscript. For the local effects (inferred from PC2), we found that hourly local $O_3$ and $NO_x$ are anticorrelated (Fig. 2, below), as expected from a chemical interaction (we did not show this figure since our primary focus was on the regional component).

[Figure]

**Figure 2: The season averaged hourly local $O_3$ and $NO_x$. Error bars represent the 95% confidence interval for the mean.**

*Referee comment*

NOx is defined in the introduction to be the sum of NO and NO2. But the monitoring NOx reported here is from chemiluminescence detectors with a molybdenum converter that also detects PAN and some HNO3. This limitation isn't critical for measurements in urban regions dominated by fresh emissions, but the contributions from PAN, HNO3, and other oxidized reactive nitrogen compounds is likely substantial if background locations and times are considered. The meaning of monitoring NOx has been discussed in many papers (e.g. , Winer et al., Response of Commercial Chemiluminescent NONO, Analyzers to Other Nitrogen-Containing Compounds, ES&T, 1974), and it should be considered here. If monitoring NOx is used to examine background levels, there needs to be considerably more examination of the data, and it may be impossible to use NOx for this sort of analysis. For example, all of the trends in NOx could be dominated by changes in partitioning between the NOx oxidation products (PAN, organic nitrates, and HNO3), rather than a reduction in NOx. If the ratio of organic nitrogen to HNO3 has changed in the background air (which is likely), then the monitoring NOx instruments would likely respond in a way that would alter trends in NOx.

Indeed the monitored total $NO_x$ might account for other oxidation products, such as PAN and $HNO_3$. However, the majority of the sites used to derive background $NO_x$ (constrained by MDA8 $O_3$) are urban sites or sites that are affected by fresh emissions; as the reviewer points out, the limitation of the method used to monitor total $NO_x$ is not a problem for these sites. Therefore, conversion to PAN and $HNO_3$ might have a weak effect on the temporal trends in background $NO_x$ in the HGB region.

There is no evidence for the significance of PAN and $HNO_3$ in the long-term measured $NO_x$ by TCEQ (5 seconds measurements averaged over 1 hour). We think that on the 1-h basis, loss of $NO_x$ by conversion to PAN and $HNO_3$ might be important in dry and sunny conditions, since the lifetime of surface PAN against photolysis and chemical losses is about 3 h, while that of $HNO_3$ against dry deposition is around 14 h. However, on the 8-h average basis, the importance of $NO_x$ conversion to PAN is reduced, because PAN has time to convert back to $NO_x$. Rapid wet deposition of $HNO_3$ during rainy days has the effects to reduce $NO_x$. Conversion of $HNO_3$ to particles can be assumed to be negligible during summer. Unfortunately, we cannot test the long-term effects of $HNO_3$ and PAN on measured $NO_x$ because no coincident $HNO_3$, PAN and precipitation data are available. A future study might look on the long-term effect of $NO_x$ converison to PAN and $HNO_3$ using 1-h solar radiation to separate between dry and wet or cloudy periods. Coincident precipitation data are not available.

The monitored $NO_x$ is the best metric that we could use for determining long-term background $NO_x$ in the HGB region. Using chemiluminescence-based NO only, would have made the separation of the regional background from the local contribution even more difficult, since NO would be more an indicative of rapid chemistry. On the other hand, the $NO_2$ reported by TCEQ is calculated as the difference between total $NO_x$ and NO. It is not really a measured value and also accounts for other oxidation products like the monitored total $NO_x$. Modeling and satellite-based $NO_2$ might be used in the future to test our data-driven background $NO_x$ in the HGB region.

Therefore, we added the following on P4, L26:

"Public data, representing 1-h average of surface measurements of $O_3$, $NO_x$ and meteorology (WD, WS and T), were downloaded from the Texas Air Monitoring and Information System website owned by TCEQ (see Data availability). The measurements were taken every five seconds and averaged over one hour. Note that, due to the measurement method (combined chemiluminescence detection-molybdenum conversion), the monitored total $NO_x$ might include traces of other oxidation products (PAN, $HNO_3$, etc.)."

We also made the following revision on P16, L17:

"Background $NO_x$ also declined in all approaches, with significant slopes (see Table 7). No other long-term background $NO_x$ studies exist, making comparison impossible. Additionally, there is no long-term evidence on the effect of $NO_x$ conversion to PAN and $HNO_3$ that could affect its temporal decline. Considering that the majority of the sites used to derive background $NO_x$ are urban sites or sites that are affected by fresh emissions, we could assume that conversion to PAN and $HNO_3$ might have had a minor effect on the temporal trends in background $NO_x$."

*Referee comment*

The background NOx value of 6.8 ppbv is surprisingly large, and it is inconsistent with the 2000 and 2006 intensive field studies that showed NOx upwind of HGB was often <1 ppbv, and NOy was a 1-4 ppbv (see for example the upwind or non-plume measurements shown in Daum, P. H., et al., A comparative study of O3 formation in the Houston urban and industrial plumes during the 2000 Texas Air Quality Study, J. Geophys. Res., 108(D23), 4715, doi:10.1029/2003JD003552, 2003; Ryerson, T. B., et al. (2003), Effect of petrochemical industrial emissions of reactive alkenes and NOx on tropospheric ozone formation in Houston, Texas, J. Geophys. Res., 108(D8), 4249, doi:10.1029/2002JD003070; Neuman, J. A., et al., Relationship between photochem- ical ozone production and NOx oxidation in Houston, Texas, J. Geophys. Res., 114, D00F08, doi:10.1029/2008JD011688). I don't know whether the discrepancy is an artifact of the data or the analysis, or both. But if background NOx were truly 6.8 ppbv, then NOx emission controls in HGB would need to be reconsidered. NOx is short-lived, and it is possible that the NOx measured at these monitoring stations is strongly influenced by local emissions. I recommend removing the analysis of background NOx, or adding substantial discussion and examination of the NOx data.

*Author response*

The two intensive field campaigns indeed show smaller $NO_x$ mixing ratios upwind the HGB region. However, they focused on very short term (1-13 days) and did not capture multi-year, multi-months and intra-seasonal variations. Moreover, these low upwind or non-plume $NO_x$ mixing ratios are measurements made from aircrafts that cannot be directly compared to background $NO_x$ derived from ground monitoring data. For example, Ryerson et al. (2003) report low non-plume mixing ratios of airborne NO and $NO_2$ (<1 ppbv), measured downwind from relevant sources (power plants and petrochemical facilities), after 6 pm. Our daytime background $NO_x$ is derived from surface measurements between 10 am and 6 pm, during May-October. Even if we ignore the different altitudes and periods of observations, the diurnal sampling is different between the two studies. Daum et al. (2003) reported low upwind $NO_x$ mixing ratios (<1 – 5 ppb) from the morning flights over the southeast of Houston, but $NO_x$ also reached 10 ppb over the city during that time. Neuman et al. (2009) measured in-plume $NO_x$ during daytime (3-5 pm) ranging from 1 to 10 ppbv. Although our daytime background $NO_x$ estimate falls within this range, a direct comparison is not possible due to inconsistent time-scales and altitudes of observations. It is possible that vertical mixing allows for significant dilution of surface $NO_x$, resulting in lower airbone mixing ratios, particularly during daytime, when the boundary layer is higher. Significant local chemistry near the surface may also contribute to reduced $NO_x$. Our first time estimate of long-term 8-h average background $NO_x$ (6.8. ppb) appears to be large compared to the hourly median approach, particularly from July to September (by 1-2 ppb). Indeed the monitoring sites are influenced by local emissions. They are also influenced by local chemistry as well as local and regional transport. By co-varying chemistry and meteorology, the PCA method could separate between the local and regional effects. We determined average regional background $NO_x$ in the HGB region only from the component identified as being "regional". We acknowledge that by constraining the 8-h average $NO_x$ and meteorology by the MDA8 $O_3$ might not be the best approach when local chemistry is important. Future studies should consider refining this estimate by analyzing the 8-h average $NO_x$, $O_3$ and meteorology that

are not constrained by MDA8 $O_3$ and see how it compares to our estimate. Considering the above, it is not justified to remove our background $NO_x$ analysis from the study.

Therefore, we propose to add the following on P1, L22 (Abstract):

"Average background $O_3$ is consistent with previous studies and between the approaches used in this study, although the approaches based on 8-h averages likely overestimate background $O_3$ compared to the hourly median approach by 7-9 ppb. Similarly, average background $NO_x$ is consistent between approaches in this study (A-C), but overestimated compared to the hourly approach by 1 ppb, on average. It is possible that we likely overestimate both background $O_3$ and $NO_x$ when the 8-h average $NO_x$ and meteorology coinciding with MDA8 $O_3$ are used in the analysis. "

Another addition would be on P17, L10 (Conclusions):

"Our estimates of 8-h based average background $O_3$ and $NO_x$ are both slightly overestimated compared to the hourly median approach, likely due to constraining the 8-h average $NO_x$ (and meteorology) by the MDA8 $O_3$. Future studies might consider refining these estimates by analyzing the 8-h average $NO_x$, $O_3$ and meteorology that are not constrained by MDA8 $O_3$."

Consequently, we also modified the previous statement on P17, L10:

"To test the linearity of the temporal trends in background $O_3$ and $NO_x$ and to continuously determine the effectiveness of control measures, and identify regulatory changes that need to be made, new studies should extend the trends in this study into future years. Additionally, wherever VOCs data are available, the extraction of background $O_3$ and $NO_x$ should be constrained over that period by VOCs as well and possibly by solar radiation. The related temporal trends should be compared over that period with those estimated from this study to highlight the effect of including VOCs and an additional meteorological variable in the multivariate analysis."

*Referee comment*

The use of MDA8 needs to be put into context, and the importance of MDA8 should be discussed. MDA8 is a regulatory construct. Is HGB in exceedance of the O3 standard? What is the current O3 standard (only the old standard is mentioned)? It would be helpful to indicate the NAAQS on the figures. The background fraction of total ozone discussed in section 3.7 also has me confused, and I think it misses the point of MDA8. The background MDA8 is important insofar as it contributes to the design value for the entire air basin. So background MDA8 should be compared with the largest MDA8 in the region to understand the effect of the background on compliance with O3 regulation. I don't see the point of comparing background MDA8 to an average of MDA8 from the same locations, as shown in figures S18-25. If the analysis finds sites and conditions that faithfully represent the background, then shouldn't the background MDA8 always equal the measured MDA8? Why are there so many points in the supplementary figures with the PCA-derived background O3 greater than the measured O3?

*Author response*

We agree that MDA8 O$_3$ is a regulatory concept and we acknowledge its importance, but it was not our goal to test if the MDA8 O$_3$ is in compliance or not. The goal of our study was to determine long-term regional background O$_3$ and NO$_x$ in the HGB. We only used MDA8 O$_3$ to separate the regional contribution to it and to better quantify its temporal trend using long term measurements and a different analytical approach compared to previous studies. The current NAAQS standard for O$_3$ is 70 ppb and our average background O$_3$ represents about 64-67% of it.

In order to quantify the contribution of the regional component to MDA8 O$_3$ in the HGB, it is well justified to compare the average background O$_3$ with the season-scale MDA8 O$_3$ averaged from the sites used to determine the background. If we were only to compare the average background with a single site showing the highest MDA8 O$_3$ we would have biased the design value for the "entire air basin". Using the highest MDA8 O$_3$ to quantify regional contributions, would also bias the design value for the entire season.

MDA8 O$_3$ at each of the "background" sites does not always equal background MDA8, unless those sites are remote, rural or relatively clean sites. The 5-10 sites used to extract background O$_3$ from MDA8 O$_3$ are all within Harris County, except for Conroe Relocated. We do not think that a single site should be decisive about the design value in the "entire air basin" and for the entire season, particularly if that single site is subjected to unexpected local emissions (i.e., wildfires).

There are several instances (all below 35 ppb) of PCA-derived background O$_3$ greater than average measured O$_3$ for the hourly median approach only (Fig. S18). This is also the case for background NO$_x$ in Fig. S19. The reason could be the intra-seasonal variation, spring versus summer/fall. We explained that for NO$_x$ at the end of section 3.7.

*Referee comment*

Some of the language is imprecise, and I had to read the sentences many times to distinguish the literal meaning from the authors' likely intent. For example, pg 2 line 14 states that "no study has yet to quantify the regional contributions to direct O3 precursors themselves: : :". I'm not sure what this means. Zhang et al, and many other papers, examines background O3 precursors, and is already referenced. The second sentence of the abstract states that ozone dependence on VOC:NOx ratio makes ozone difficult to control locally. I think the whole point of this paper is that large background contributions, rather than the VOC:NOx, may make local ozone control challenging. I don't understand page 3, line 1 that says "Meteorological controls : : :are reflected by a more significant decline .. in the east than in the west". Cooper et al explain this difference by changes in Asian emissions and biomass burning, not changes in meteorology. I don't understand page9, line 19: why does NOx increase with windspeed? The last paragraph of the conclusion is not supported by the manuscript. Rather than emphasize work that needs to be performed, the authors should focus on their most important findings.

*Author response*

For clarity, we rephrased P2, L14-16 as:

"No long-term study exists that quantifies the regional contributions to direct $O_3$ precursors themselves, such as nitrogen oxides ($NO_x$ = nitrogen dioxide ($NO_2$) + nitric oxide ($NO$)). Our goal is to better characterize the trends in regional background $O_3$ and $NO_x$ in the HGB region on the decadal scale."

Regarding the ozone dependence on VOC/$NO_x$ ratio, we rephrased line 2-3 in the Abstract:

"Because of its dependence on the volatile organic compounds (VOCs) to nitrogen oxides ($NO_x$) ratio, ground-level $O_3$ is difficult to control locally, where many sources of these precursors contribute to its mixing ratio."

to read:

"Local precursor emissions, such as volatile organic compounds (VOCs) and nitrogen oxides ($NO_x$), together with their chemistry contribute to the $O_3$ and $NO_x$ mixing ratios in the HGB region."

P3, L1: It is our interpretation at the scale of US. We did not say that meteorology changes, we did imply that the meteorological controls are different in the west than in the east and they are reflected into a differential decline of $O_3$ at the scale of the US. For clarity, we rephrase this line on page 3:

"Meteorological controls on the scale of the US also may play a role in the differential decline during recent decades of summer surface $O_3$ observed in the east, southeast and midwest (Cooper et al., 2012; Hudman et al., 2009) than in the west (Cooper et al., 2012). There are different meteorological controls in the west (i.e., thermal inversion and orographic lifting, Langford et al., 2010), which can either increase $O_3$ locally or transport $O_3$ up in the free troposphere and towards east. Additionally, the pollution transport from Asia contributes to a higher $O_3$ in the western US compared to the eastern US (Cooper et al., 2012)."

P9, L19: Fig. S6e shows that at wind speeds > 4 m s$^{-1}$, the PC2-$NO_x$ scores are positive (suggesting an increase in $NO_x$). This is the case for October. During this month, winds were from SE (Fig. S6d). Together the two wind variables say that some regional $NO_x$ is also included in the second component during this month, on the 1-h basis.

P17, L10: Our study opens the paths for new research. Therefore, it is important to point out how future studies should be focused. The last paragraph is the most appropriate for this purpose. However, we modified the conclusions (see our answers above related to P17, L10).

*Referee comment*

The results reported here can be made more valuable by further synthesizing the findings. There are 25 figures in the Supplementary section, and it is hard to distinguish one from the next. The first 6 tables are very dense, showing many PCA loadings for many different sites. I don't think many readers will be able to use all these tables of numbers and all the figures in the Supplementary. This paper examines many topics, and most points are supported by a scatter plot and the associated statistics obtained from a linear least squares fit. It is challenging to appreciate the important findings, as they are obscured by an abundance of data and statistics. An in-depth consideration of a single topic, such as the decadal change in wind direction and its effect on background ozone, would be a more powerful contribution to the literature.

*Author response*

Regarding the tables, we reduced them to five, thus keeping only the tables containing the loadings because they were so important for interpreting the meaning of the principal components. Therefore, we propose to remove Tables 1 and 4 as they appeared in the manuscript and replace the current Table 3 by that containing the loadings for Approach A.

By just focusing on a single topic as suggested "wind direction and its effect on background ozone, would be a more powerful contribution to the literature" we would limit our study to what others did. We wanted to look at the data in different ways to improve the estimation of regional background $O_3$ in the HGB region on the longest term possible and to assess its trends. To estimate regional background $NO_x$ it was important to analyze $O_3$ and $NO_x$ simultaneously. We added one more level of complexity by simultaneously analyzing chemistry and meteorology. These are not different topics. In this context, we found it important to report all the relevant statistics that provide support to the figures and to our analysis and interpretation.

P7, L16: Deleted "Table 1 summarizes the retained components, along with the fraction of variance explained by each of them."

P7, L22: Replaced "Table 2" by "Table 1"

P9, L2: Deleted "(Table 1)"

P10, L11: Deleted the comma after "The former"

P11, L12: Added "; their loadings are shown in Table 2"

P12, L4: Deleted ", as summarized in Table 4"

P12, L7: Replaced "Table 5" by "Table 3"

P12, L31: Deleted "(Table 5)"

P12, L31: Removed the entire paragraph and the related Figures from the SI (Figs. S14-S15). These plots were not necessary since the variables were not determined independently in approach B.

P13, L13: Replaced "Table 6" by "Table 4"

P13, L15: Removed the entire paragraph and the related Figures from the SI (Figs. S16-S17). These plots were not necessary since the variables were not determined independently in approach C.

P13, L26: Added "vs."

P13, L30: Rephrased the sentence to: "The three approaches (A-C) yield similar values for July, when local chemistry is expected to be more important (Nielsen-Gammon et al., 2005)."

P13, L31: Rephrased the sentence to: "The sudden increase from July to August is consistent in all approaches (significant regional summertime chemistry), but background $O_3$ starts decreasing earlier for Approaches B and C compared to the

hourly median and Approach A, likely the result of changes in meteorology after August (less influence from sea breeze effects).”

P14, L23: Rephrased the sentence to sound “Compared to the SE wind-constrained slopes from Berlin et al. (-0.92 ± 0.74 ppb y$^{-1}$ or -0.79 ± 0.65 ppb y$^{-1}$), our slope is much smaller but closer to that from Souri et al. (0.09 ± 0.40 ppb y$^{-1}$).”

P14, L24: Rephrased the sentence to: “The mean background $O_3$ over the seventeen years is 46.74 ± 0.58 ppb and compares well with the 14-y and 15-y means from  Berlin et al. (2013) and Souri et al. (2016) (42.5 ± 6.3 ppb and 57 ± 19 ppb respectively), representing SE influences only.”

P14, L26: Deleted “(2013)”

P15, L3: Corrected “ppb/yr” to “ppb y$^{-1}$”

P15, L4: Rephrased and extended the sentence: “Relative to a previous study (Berlin et al., 2013), the slope is less steep (-0.68 vs. -0.92 ppb y$^{-1}$ or -0.79 ppb y$^{-1}$), but its error is halved (42% vs. 80%, respectively). Our slope, though smaller, compares well in terms of absolute error with the slope from Souri et al. (2016), describing continental regional background $O_3$ (-1.0 ± 0.55 ppb y$^{-1}$); however, as Souri et al. suggested, local sources may have contributed half to the observed $O_3$ within the E-NE wind cluster, which could explain the steeper slope observed in their study. They also reported a weaker slope for regional background $O_3$ from the E-SE (-0.9 ± 0.86 ppb y$^{-1}$).”

P15, L5: Corrected the sentence to start: “As observed in Fig. 6, a”

P15, L7: Rephrased the sentence to read: “Also, State of Texas controls on precursor emissions implemented in 2007 (Berlin et al., 2013) may also have contributed to reduced background $O_3$ after that.”

P15, L9: Corrected “versus” by “vs.”

P15, L22: Corrected “(Fig. S18 to Fig. S25)” to read “(Fig. S14 to Fig. S21)

P16, L2: Corrected “(Fig. S19)” to read “(Fig. S15)

P16, L6: Added “vs.” and deleted the comma after “summer/fall”

P16, L11: Replaced “Table 7” by “Table 5”

P16, L11: Added “and Souri et al. (2016)”

P16, L11: Corrected slope value to read “-0.68 ± 0.27 ppb y$^{-1}$”

P16, L15: Corrected slope value to read “-0.68 ± 0.27 ppb y$^{-1}$”

P16, L15: Added “Overall, the slopes from different approaches in this study and other studies are not significantly different (Fig. 8).”

P16, L15: Rephrased the sentence and added a new sentence: “The average background $O_3$ in this study is slightly larger (by 2-4 ppb) compared to that reported by Berlin et al. (2013), in any of the approaches except for the hourly median approach, which is smaller by 5 ppb. However, compared to Souri et al. (2016) the average estimates from our study and Berlin et al. are all much smaller, with differences ranging from 10 to 69 ppb (Table 5).“

P16, L30: Rephrased to include Souri et al.: “This is consistent with results from two previous studies (Berlin et al., 2013; Souri et al., 2016).”

P17, L7: Rephrased the sentence to read: "However, in our study, regional contributions to average MDA8 $O_3$ are underestimated when the space-time covariance of meteorology and chemistry is not considered (Fig. S16 vs. Fig. S18). When this covariance is accounted for in the analysis (our Approach B), the associated temporal trend in background $O_3$ (or $NO_x$) reflects both the effects of controlling precursor emissions and changes in meteorology. For instance, local chemistry was much more important in earlier years (prior to 2007) due to high emissions of $O_3$ precursors from petrochemical facilities, making it difficult to extract the regional background from surface data during those years. The trend became steadier after 2007 probably as an effect of emissions controls and a prevailing S-SE flow; this latter is consistent with the observed increased frequency of the southerly flow from the GOM (Liu et al., 2015). Based on a previous study (Wang et al., 2016), variations in the intensity and location of the Bermuda High could also explain some of the temporal behavior in summertime MDA8 $O_3$, causing a drop in mid-July, when southerly flow from the GOM is allowed to enter the region; this is marine background $O_3$ and also contributes to the decline in regional background $O_3$ over time. We also observed this effect in regional background $O_3$ during July, particularly when using the hourly median approach."

P17, L14: Added "Coincident solar radiation and $NO_x$ could also be used to test the conversion of $NO_x$ to oxidation products (PAN, $HNO_3$, etc.) and asses the magnitude of this effect on the declining background $NO_x$ in the HGB region."

**Changes in the Supplement Information (SI):**

P1: Added title: Supplement Information

P11 to P12: Removed Figures S14 to S17

P13: Renumbered "Figure S18" as "Figure S14" and "Figure S19" as "Figure S15"

P14: Renumbered "Figure S20" as "Figure S16" and "Figure S21" as "Figure S17"

P15: Renumbered "Figure S22" as "Figure S18" and "Figure S23" as "Figure S19"

P16: Renumbered "Figure S24" as "Figure S20" and "Figure S25" as "Figure S21"

[revised manuscript text omitted]

---

## Author Response (AR2)

**Author response to co-Editor's decision "Reconsider after major revisions"**

We appreciate the careful attention that the reviewers and editor have given to our work and have done our best below to address their concerns. A number of the comments we received describe three issues:

1. Potential biases in 8-h background $NO_x$ estimates (either due to time averaging or overdetection of $NO_x$)
2. Overestimation, comparison with aircraft data and representativeness of background $NO_x$
3. Comparison of 8-h background $O_3$ with MDA8 $O_3$ on exceedance days

For each issue, we address the comments first with an overview of the issue and continuing with specific changes in the revised manuscript. Each comment is addressed in the same order as the original editor's report below, with specific corrections listed. In addition, we provide a list of other changes at the end of this author response, followed by the revised manuscript, including the tracked changes.

We have performed a thorough bias assessment individually for each of these potential issues (1, 2) and determined that they lead to not more than a 35% overestimation of background $NO_x$ (~ 2 ppb). To avoid adding length to the manuscript, we have added text describing this bias analysis in the SI.

1. Potential biases in 8-h background $NO_x$ estimates (either due to time averaging or overdetection of $NO_x$)
This overall concern occurs through a number of editor's comments (1-3). We separately address below the two potential biases.

    a. Bias due to time-averaging

*Comment 1 (background $NO_x$ averaging)*: R2 has requested more information to assess the analysis of background $NO_x$. R2 asked that evidence be given to whether the time averaging adequately captures background $NO_x$. The response that no other studies exist with which to compare long-term background $NO_x$ studies is not sufficient. The referenced short-term studies are not useful as these report tens of ppb $NO_x$, so they must be referring to something other than background $NO_x$. Figure 2 shows evidence that local effects are important in PC2, but could this indicate an issue with the MDA8 $NO_x$ metric generally? Have the authors looked at $O_x$ (= $NO_2 + O_3$), this would remove the variability due to $O_3$ and $NO_x$ changes due to storage of $O_3$ as $NO_2$.

*Author response*: The editor and reviewer are concerned that our 8-h averaging of $NO_x$ does not adequately capture background $NO_x$. This could occur because the temporal variability in $NO_x$ and $O_3$ are different. For the

determination of annual temporal trends in background $O_3$ and $NO_x$ we had to choose one daily 8-h average value in order to use both variables in the statistical analysis. We chose MDA8 $O_3$ and this forced our choice of the corresponding 8-h $NO_x$ value. However, because the temporal variability in $NO_x$ and $O_3$ are different, this choice may introduce a bias into our $NO_x$ background values (possibly an overestimation given the choice of MDA8 $O_3$ and the 8-h averaging of $NO_x$).

We address the time-averaging bias in section 1.1 (p. 15, SI), directly below. We address the potential use of the $O_x$ metric in section 1b of this author response (see the quoted section 1.2).

**"1. Considerations for potential biases in background $NO_x$**

Background $NO_x$ estimates in this study may be subject to two biases: the time-averaging scale and the overdetection of $NO_x$ by the measuring instrument. Both biases lead to overestimation of background $NO_x$. Thus, our estimate provided an upper bound of background $NO_x$. In the following sections we discuss, estimate and use these biases to quantify the long-term averages and temporal trends in lower bound background $NO_x$. Because we subtract these biases from the previously determined background $NO_x$ (*aka* upper bound), our new estimate of background $NO_x$ is a lower bound.

**1.1 Bias due to time-averaging**

The 8-h averaging of $NO_x$ may not adequately capture background $NO_x$. This could occur because the temporal variability in $NO_x$ and $O_3$ are different. For our PCA approach in which we co-varied chemistry and meteorology, it was necessary that the averaging of $NO_x$, WS, WD and T occur over the same time period as for MDA8 $O_3$ to match the analysis time scale for all the variables at which their covariance would be meaningful. For instance, $O_3$ and T vary over larger temporal scales compared to $NO_x$ and WS. However, because $O_3$ and $NO_x$ have different temporal dynamics, this may introduce bias. In our study, we selected the maximum 8-h average $O_3$ value which, therefore, drove our selection of the corresponding 8-h average $NO_x$, WD, WS and T values, potentially biasing high our background $NO_x$ estimate and its temporal trends.

To determine how much we may have overestimated background $NO_x$ due to time-averaging, we have compared the season-scale 8-h and 1-h background $NO_x$ values (Fig. 5). We found that due to differing temporal dynamics of $O_3$ and $NO_x$ the analytical focus on the time of MDA8 $O_3$ occurrence may lead to an overestimation of the 8-h background $NO_x$ of approximately 18%."

b. Bias due to overdetection of $NO_x$ by the instrument

*Comment 2 (background $NO_x$ measurements):* This comment has not been addressed. If $NO_x$ is averaged over the same time period as MDA8 then it is most susceptible to interferences by higher nitrogen oxides, as the MDA8 typically centers over the hours of active photochemistry. R2 is asking whether the trend in NOx is in part driven by chemiluminescence instruments having different sensitivity to organic nitrates than to $HNO_3$. Under high-$NO_x$ chemistry, $HNO_3$ is the dominant $NO_x$ termination product, but at intermediate and low $NO_x$, organic nitrate production is more important. If the chemiluminescence instruments are more sensitive to one type of nitrate over another, the trend in observed $NO_x$ may be impacted. The authors should present evidence to address the concern raised by R2. The relevance of the discussion of the $HNO_3$ lifetime is not clear to me.

*Author response:* The editor and reviewer are concerned that instrument bias may lead to overdetection of $NO_x$. The $NO_x$ analyzer also detects other reactive nitrogen ($NO_y$) species (PAN, $HNO_3$, organic nitrates, etc.) as $NO_x$. In other words, the decline in background $NO_x$ might be somewhat explained by the detection of $NO_2$ stored as another $NO_y$ species and that this bias varies with the $NO_x$ regime (high vs. low/intermediate).

We address this bias in section 1.2 (p. 15, SI), directly below:

**"1.2 Bias due to overdetection of $NO_x$ by the measuring instrument**

The combined chemiluminescence/molybdenum conversion method used to measure $NO_x$ also detects other reactive nitrogen ($NO_y$) species (PAN, $HNO_3$, organic nitrates/nitrites, etc.) as $NO_x$ (Winer et al., 1974). Because $NO_2$ is determined by difference from total $NO_x$ (after its conversion to NO) and NO (measured by chemiluminescene prior to the conversion of other $NO_y$ species to NO), $NO_2$ is actually not measured by this method, preventing us from using it in the analysis of background $NO_x$ or using the $O_x$ metric.

Although there is a variation in the instrument conversion efficiency for individual species, we assume 100% conversion for all species to generate an upper-bound estimate of error, allowing us to use published field studies to determine the potential error in overdetecting $NO_x$. A two-month field study (Luke et al., 2010) individually measured $NO_x$ species (NO, $NO_2$) and $NO_z$ species ($HNO_3$, PAN, HONO and p-$NO_3^-$) during daytime at the top of the Moody Tower (University of Houston). The dominant $NO_z$ species, representing fractions of $NO_y$ during the most photochemically active window (11 am - 5 pm), were $HNO_3$ (17-19.5 %) and PAN (12-14.8%), while other species (HONO and p-$NO_3^-$) showed minor contributions (<1-1.5%). Based on the total $NO_z$ fraction and assuming that $NO_z$ would

be detected as $NO_x$ in the TCEQ instruments, we inferred that the overdetection of $NO_x$ by the chemiluminescense/molybdenum conversion method was at most approximatively 30%. This estimate is likely to be high for two reasons. First, as described above, instrument conversion of PAN, $HNO_3$ and higher nitrogen oxides is not likely to be 100%. Second, the extrapolation of this error estimate to the season-scale (May-Oct.) includes periods of time when photochemistry is not as active as the summer, leading to less production of $NO_z$ species."

Finally, we acknowledge the corrections of background $NO_x$ for the overall bias in section 1.3 (p. 16, SI), directly below:

**"1.3 Corrections of background $NO_x$ for overall bias**

Based on these two potential errors and using error propagation, we estimated a maximum overall bias of 35%. By applying this overall bias to the upper bound of background $NO_x$ (the one uncorrected for biases), we obtained a lower estimate of the background, which we refer to as the lower bound of 8-h background $NO_x$ (see Table 5). Using the bias due to the measurement method only, we estimated the lower bound of 1-h background $NO_x$ (Table 5). Given that the bias due to overdetection of $NO_x$ was based on late summer only, a period highly representative of significant photochemistry, we hypothesize that the actual season-scale (May-Oct.) ground-level background $NO_x$ falls in between these two bounds. The lower bound of background $NO_x$ is shifted to lower ranges compared to the uncorrected values by approximately 2 ppb. Likewise, the regression coefficients (slope and intercept) decreased proportional to the overall bias (by ca. 35%). Figure S22 shows the lower bound of 8-h background $NO_x$ trends for all approaches (A-C). The long-term averages and trend parameters for these estimates are summarized in Table 5."

[Figure]

Approach A ($R^2 = 0.53$) = (-0.039 ± 0.019)t + (82 ± 38)
Approach B ($R^2 = 0.58$) = (-0.026 ± 0.011)t + (57 ± 23)
Approach C ($R^2 = 0.30$) = (-0.009 ± 0.008)t + (21 ± 15)

△ Approach A (LB)   ● Approach B (LB)   ◇ Approach C (LB)

**Figure S22: Temporal trends in lower bound of background $NO_x$**

For line-by-line revisions of the manuscript acknowledging the biases and the lower bound estimates of background $NO_x$, please see section 4 of this author response.

2. Overestimation, comparison with aircraft data and representativeness of background $NO_x$

The overestimation of 8-h background $NO_x$ has been addressed in section 1b above. However, we refer to it briefly in our response to Comment 3 below, which has two additional points: comparison of background $NO_x$ with aircraft data and representativeness of our background $NO_x$ as "background $NO_x$".

*Comment 3 (background $NO_x$ is large)*: More should be done here. While the aircraft studies are carried out over a short time period, they are still informative. Can the results be compared over the same time window? Second, due to concerns over interferences by higher oxides, high background $NO_x$ deserves special attention. Ozone is produced throughout the boundary layer, if the background $NO_x$ estimate is influenced by very-local sources, then it is not representative of background $NO_x$, including vertically.

*Author response*: The editor and reviewer are primarily concerned that our background $NO_x$ estimates are too large and that this could possibly be related to higher oxides (e.g., $HNO_3$) interfering with the $NO_x$ measurements. We discussed this bias above and corrected background $NO_x$ for the overall bias (see section 1b). An additional concern is whether our long-term average background $NO_x$ (which appears to be too large) could be compared to short-term aircraft measurements of $NO_x$ over the same daytime window. Aircraft studies are important for understanding rapid plume chemistry and downwind effects; they are very informative because the conditions near the ground differ from those up in the boundary layer and at various times during the high $O_3$ season. However, our results are not directly comparable over the same time window because the aircraft measurements are taken at intervals of seconds and are not representative for the multi-year, 6-month observations from which we derived the 1-h median or the 8-h average daily values. We did not use the 1-s or 5-m ground-monitored data in any of the approaches. Moreover, a direct comparison between the ground-level background $NO_x$ and upper level 'non-plume' $NO_x$ over the same daytime window might be reasonable only if the ground-level value is conserved during vertical updraft and that the higher altitude background air does not change. Neuman et al. (2009) pointed out that background air varied temporally and spatially due to variable and higher wind speeds. Despite the fact that we cannot directly compare our background $NO_x$ with the aircraft data, we still mention aircraft values in one line-by-line revision of the manuscript (see P19, L26 in section 4 of this author response).

Finally, the editor suggests that local influences may impede a proper estimation of background $NO_x$ and so our background $NO_x$ may not be representative of the actual background $NO_x$ either near the ground or vertically. We agree that local influences may bias our background $NO_x$ estimates, but this bias was accounted for by the PCA method, which helped to separate between regional and local contributions. Biases caused by temporal averages are discussed elsewhere (see section 1a above).

Our background $NO_x$ is representative for the ground-level, because it was derived from ground-monitoring data. As such, it is expected that background $NO_x$ is not representative vertically, given its smaller temporal variability, while $O_3$ could be since it is formed throughout the boundary layer and varies over larger time-scales than $NO_x$. However, because we used ground-level data, we consider that both $O_3$ and $NO_x$ backgrounds are representative for the ground level only.

To clarify terminology regarding background $O_3$ and $NO_x$, we added the following on P19, L2:

"Because we used ground-monitoring data, both background $O_3$ and $NO_x$ determined in this study represent the ground-level backgrounds, describing influences from regional chemistry and transport."

We also rephrased the sentence on P1, L14:

> "In this study, we estimate ground-level regional background $O_3$ and $NO_x$ in the HGB region and quantify their decadal-scale trends."

3. Comparison of 8-h background $O_3$ with MDA8 $O_3$ on exceedance days

*Comment 4 (MDA8 $O_3$)*: I agree with the referee that MDA8 use needs context. The fraction of $O_3$ on all days attributed to the background is not important, what is important is the contribution to MDA8 on exceedance days. I do not understand why the authors have chosen the MDA8 metric but are not concerned with compliance/exceedances. As acknowledged in the $NO_x$ discussion, the MDA8 may not be an ideal averaging window for assessment of background $NO_x$ concentrations. If the analysis aims to assess trends in background $O_3$ and $NO_x$ outside of the regulatory context, then perhaps there is a better metric.

*Author response:* The editor and reviewer are concerned that we did not consider the 8-h background $O_3$ in the context of MDA8 $O_3$, which is a regulatory concept, because we did not focus on days when MDA8 $O_3$ exceeded the current NAAQS value of 70 ppb. Moreover, the time-averaging imposed by the use of the MDA8 $O_3$ metric, partially triggered the overestimation of background $NO_x$. In this context, the editor suggests that maybe there are other metrics that could be used to estimate trends in background $O_3$ and $NO_x$ outside the regulatory concept. We address these concerns below.

We respectfully disagree about background only mattering on exceedance days, if there is a non-zero threshold of health impact of $O_3$ or if standards change. We used MDA8 $O_3$ because previous studies used it, and we wanted to be able to compare our results in terms of regional background $O_3$ and its temporal trend, particularly because we used an innovative approach (i.e., covariance of chemistry and meteorology). We also know from previous studies (Nielsen-Gammon et al., 2005) that regional background is an important fraction of MDA8 $O_3$. An additional goal was to quantify this fraction using various long-term approaches based on continuous data that are relevant at the temporal scale of the high $O_3$ season and at the spatial scale of the studied area. Moreover, regional contributions to season-scale and area-averaged MDA8 $O_3$ for all days could be useful in modeling studies.

We have discussed the overestimations of background $NO_x$ due to time-averaging and corrected it for the overall bias (see section 1 above).

Regarding the use of a metric outside the regulatory concept, we can only think of $O_x$ ($O_3+NO_2$), but this was not possible in our study because $NO_2$ is not actually measured by the TCEQ instrument.

4. Other changes in the previously revised manuscript:

*a. References*: We added two more refererences:

Luke, W. T., Kelly, P., Lefer, B.L., Flynn, J., Rappenglück, B., Leuchner, M., Dibb, J., E., Ziemba, L. D., Anderson, C. H., Buhr, M.: Measurements of primary trace gases and NOy composition in Houston, Texas, Atmos. Environ., 44, 4068-4080, doi:10.1016/j.atmosenv.2009.08.014, 2010.

Winer, A. M., Peters, J. W., Smith, J. P., Pitts, J. N. Jr.: Response of commercial chemiluminescent NO-$NO_2$ analyzers to other nitrogen-containing compounds, Environ. Sci. Technol., 8(13), doi: 10.1021/es60098a004, 1974.

*b. Tables and Figures*

Table 5: We added values for the lower bound of background $NO_x$

Fig. 7: We modified the caption to include "upper bound"

*c. Text revisions in the manuscript*

P6, L8. We corrected the sentence to read:

"The measurements were taken every second, averaged over five minutes and then averaged over one hour."

P15, L1-3. We rearranged and rephrased the two sentences for ease of reading:

"The simultaneous effect of increasing the spatial scale and reducing the temporal scale of the analysis (constrained by the availability of continuous data) was studied using Approach C. Therefore, results in this section were driven by the use of five more sites and a shorter study period compared to Approach B."

P16, L10. We added the sentence:

"In addition, we wanted to assess the effects of co-varying chemistry and meteorology on these trends."

P16, L24. We added the following text to acknowledge the integration of biases in background $NO_x$:

"Note that due to potential biases in background $NO_x$ (p.15-16 in the SI), this value represents the upper bound in background $NO_x$. After taking into account the overall bias, we also estimated a lower bound in background $NO_x$ of $4.49 \pm 0.12$ ppb (see Table 5 for all approaches). The linear trends for all approaches were shifted to lower ranges by ca. 2 ppb, on average (Fig. S22)."

P17, L9. We rephrased the sentence and added a new sentence:

"The 17-y mean of background $NO_x$ ($6.80 \pm 0.13$ ppb), representing the upper bound, is in good agreement with Approach A. The average value corresponding to the lower bound of background $NO_x$ is $4.45 \pm 0.08$ ppb."

P17, L16. We rephrased the sentence on this line and added a new one:

"The 13-year mean of background $O_3$ is 44.71 ± 1.28 ppb, while of mean upper bound background $NO_x$ is 6.03 ± 0.05 ppb. The lower bound estimate of mean background $NO_x$ represents 3.95 ± 0.03 ppb."

P17, L30. Added the following:

"(see p. 15-16 in the SI for potential biases)"

P18, L18. We rephrased the sentence to read:

"Both upper and lower bounds of background $NO_x$ also declined in all approaches, with significant slopes (see Table 5)."

P18, L22: We modified the sentences on these lines and added two new sentences:

"Additionally, there is no long-term and season-scale evidence on the effect of $NO_x$ conversion to PAN and $HNO_3$ that could affect its temporal decline. Considering that the majority of the sites used to derive background $NO_x$ are urban sites or sites that are affected by fresh emissions, we could assume that conversion to PAN and $HNO_3$ might have had a minor effect on the annual trends in background $NO_x$ and at the 6-months scale. However, we estimated a bias of ca. 30 % due to detection of PAN, $HNO_3$ and other nitrogen species as $NO_x$ (see p. 15-16 in the SI). This, combined with the bias due to 8-h averaging of $NO_x$, has shifted the annual trends to lower ranges by 2 ppb."

P19, L5. We revised the sentence to acknowledge the two bounds in background $NO_x$:

"Similarly, we detected and quantified a decline in the upper and lower bounds of background $NO_x$ in all approaches."

P19, L26. We rephrased the sentence and added three more sentences:

"Future studies might consider refining these estimates by using a smaller time-averaging scale for $NO_x$, $O_3$ and meteorology. Although we estimated a bias of 18% due to 8-h averaging of $NO_x$, future refinements of background $NO_x$ would probably reduce this bias. In addition, corrections of $NO_x$ measurements that are representative for the region and the time periods analyzed in this study are highly recommended to further improve the lower bound estimate of background $NO_x$; the average value of ca. 4 ppb still appears to be large compared to the short-term aircraft 'non-plume' $NO_x$ of 1-1.5 ppb observed in the region."

P1, L20. We rephrased the sentence to read:

" Likewise, the estimation of regional background $NO_x$ trend constrained by $O_3$ and meteorology was -0.04 ± 0.02 ppb $y^{-1}$ (upper bound) and -0.03 ± 0.01 ppb $y^{-1}$ (lower bound).

P1, L21. We modified the sentence to read:

"Our best estimates of 17-y average of season-scale background $O_3$ and $NO_x$ were 46.72 ± 2.08 ppb and 6.80 ± 0.13 ppb (upper bound) or 4.45 ± 0.08 ppb (lower bound), respectively."

P1, L24. We revised the sentence on this line:

"Similarly, the upper bound of average background $NO_x$ is consistent between approaches in this study (A-C), but overestimated compared to the hourly approach by 1 ppb, on average."

P1, L27. We added a new sentence:

"We likely overestimate the upper bound background $NO_x$ due to instrument overdetection of $NO_x$ and the 8-h averaging of $NO_x$ and meteorology coinciding with MDA8 $O_3$."

*d. Changes in the supplement information (SI)*

Page 15-16: We added a new section: "Background $NO_x$", which describes potential biases in background $NO_x$.

[revised manuscript text omitted]